# The interferon-inducible GTPase MxB promotes capsid disassembly and genome release of herpesviruses

**Manutea C Serrero**[1,2], **Virginie Girault**[3], **Sebastian Weigang**[4], **Todd M Greco**[5], **Ana Ramos-Nascimento**[1], **Fenja Anderson**[1], **Antonio Piras**[3], **Ana Hickford Martinez**[1], **Jonny Hertzog**[6], **Anne Binz**[1,2,7], **Anja Pohlmann**[1,2,7], **Ute Prank**[1], **Jan Rehwinkel**[6], **Rudolf Bauerfeind**[8], **Ileana M Cristea**[5], **Andreas Pichlmair**[3,9], **Georg Kochs**[4], **Beate Sodeik**[1,2,7]\*

[1]Institute of Virology, Hannover Medical School, Hannover, Germany; [2]RESIST - Cluster of Excellence, Hannover Medical School, Hannover, Germany; [3]Institute of Virology, Technical University Munich, Munich, Germany; [4]Institute of Virology, Freiburg University Medical Center, University of Freiburg, Freiburg, Germany; [5]Department of Molecular Biology, Princeton University, Princeton, United States; [6]MRC Human Immunology Unit, MRC Weatherall Institute of Molecular Medicine, Radcliffe Department of Medicine, University of Oxford, Oxford, United Kingdom; [7]German Center for Infection Research (DZIF), Hannover-Braunschweig Partner Site, Hannover, Germany; [8]Research Core Unit Laser Microscopy, Hannover Medical School, Hannover, Germany; [9]German Center for Infection Research (DZIF), Munich Partner site, Munich, Germany

**\*For correspondence:** sodeik.beate@mh-hannover.de

**Competing interest:** The authors declare that no competing interests exist.

**Abstract** Host proteins sense viral products and induce defence mechanisms, particularly in immune cells. Using cell-free assays and quantitative mass spectrometry, we determined the interactome of capsid-host protein complexes of herpes simplex virus and identified the large dynamin-like GTPase myxovirus resistance protein B (MxB) as an interferon-inducible protein interacting with capsids. Electron microscopy analyses showed that cytosols containing MxB had the remarkable capability to disassemble the icosahedral capsids of herpes simplex viruses and varicella zoster virus into flat sheets of connected triangular faces. In contrast, capsids remained intact in cytosols with MxB mutants unable to hydrolyse GTP or to dimerize. Our data suggest that MxB senses herpesviral capsids, mediates their disassembly, and thereby restricts the efficiency of nuclear targeting of incoming capsids and/or the assembly of progeny capsids. The resulting premature release of viral genomes from capsids may enhance the activation of DNA sensors, and thereby amplify the innate immune responses.

## Editor's evaluation

This paper uses an innovative cell-free protein-protein interaction system to identify factors that interact with HSV-1 capsids in infected cells. In addition to cataloging numerous capsid-interacting proteins, the manuscript probes the antiviral mechanism of one of these, MxB. The data provide strong support for an intriguing model in which MxB "punches" holes in HSV-1 capsids, releasing viral DNA and potentially triggering host DNA sensors. Moreover, the results suggest that viral proteins bind to and shield the capsids from MxB attack, offering a new perspective on how viruses might evade some host defenses.

## Introduction

Infections with human alphaherpesviruses are associated with painful and stigmatizing manifestations such as herpes labialis or herpes genitalis, but also cause life-threatening meningitis or encephalitis, potentially blinding eye infections, herpes zoster, and post-herpetic neuralgia, particularly in immunocompromised patients (*Gershon et al., 2015*; *Whitley and Roizman, 2016*; *Whitley and Johnston, 2021*). Herpes simplex viruses (HSV-1, HSV-2) and varicella zoster virus (VZV) productively infect epithelial and fibroblast cells of the skin and mucous membranes as well as neurons, but are restricted in immune cells. Macrophages, Langerhans cells, dendritic cells, and NK cells mount potent immune responses against alphaherpesviruses (*Whitley and Roizman, 2016*).

Intracellular DNA sensors are crucial to sense herpesvirus infections, and to induce caspase-1-mediated inflammation and type I IFN expression (*Hertzog and Rehwinkel, 2020*; *Kurt-Jones et al., 2017*; *Lum and Cristea, 2021*; *Ma et al., 2018*; *Paludan et al., 2019*; *Stempel et al., 2019*). During an unperturbed infection, capsid shells shield herpesviral genomes from cytosolic sensors during nuclear targeting as well as after nuclear genome packaging (*Arvin and Abendroth, 2021*; *Döhner et al., 2021*; *Knipe et al., 2021*). HSV-1 capsids can withstand compressive forces of up to 6 nN which is more than sufficient to endure the 18 atm repulsive pressure of the packaged viral DNA (*Bauer et al., 2013*; *Roos et al., 2009*). So far, it is unclear how cytosolic DNA sensors gain access to herpesviral genomes; either cytosolic host factors disassemble the sturdy herpesviral capsids during infection, or the nuclear envelopes become leaky.

HSV-1 virions contain an amorphous tegument layer that links the icosahedral capsids with a diameter of 125 nm to the viral envelope proteins (*Crump, 2018*; *Dai and Zhou, 2018*; *Diefenbach, 2015*). To identify cytosolic proteins that promote or restrict infection by interacting with HSV-1 capsids, we have developed cell-free methods to reconstitute capsid-host protein complexes using tegumented capsids from extracellular viral particles or tegument-free capsids from the nuclei of infected cells (*Radtke et al., 2014*). Intact capsids are incubated with cytosol prepared from tissues or cultured cells, and the capsid-host protein complexes are isolated, and characterized by mass spectrometry (MS), immunoblot, electron microscopy, and functional assays. We could show that HSV-1 capsids require inner tegument proteins to recruit microtubule motors, to move along microtubules, to dock at nuclear pore complexes (NPCs), to release viral genomes from capsids, and to import viral genomes into the nucleoplasm, and that capsids lacking tegument cannot move along microtubules, but still bind to nuclear pores (*Anderson et al., 2014*; *Ojala et al., 2000*; *Radtke et al., 2010*; *Wolfstein et al., 2006*).

Here, we searched for proteins that might contribute to sensing cytosolic capsids and thereby promote the detection of herpesviral genomes. Using extracts of matured THP-1 cells, a model system for human macrophages (*Tsuchiya et al., 1980*) we identified type I interferon (IFN) inducible proteins that bound specifically to HSV-1 capsids. Among them was the large dynamin-like GTPase myxovirus resistance protein B (MxB). MxB limits the infection of several herpesviruses, and can mediate almost 50% of the IFN-mediated restriction of HSV-1, although its mode of action has remained elusive so far (*Crameri et al., 2018*, *Liu et al., 2012*, *Schilling et al., 2018*, *Jaguva Vasudevan et al., 2018*). MxB has been first described for its potent inhibition of HIV infection (*Goujon et al., 2013*; *Kane et al., 2013*, *Liu et al., 2013*). The human *MX2* gene codes for a full-length MxB (residues 1–715) and a smaller version (residues 26–715) that lacks an N-terminal extension (NTE), which both are highly expressed upon IFN induction (*Melén et al., 1996*). MxB likely operates as an anti-parallel dimer but can also form higher-order filaments; its N-terminal GTPase domain connects to a bundle signaling element that moves relative to the GTPase domain in response to nucleotide binding, and the C-terminal stalk domain is critical for MxB oligomerization (*Alvarez et al., 2017*; *Chen et al., 2017*; *Fribourgh et al., 2014*; *Gao et al., 2011*).

We show here that both, full-length MxB(1-715) and MxB(26-715) have the remarkable property to disassemble the capsids of the three human alphaherpesviruses HSV-1, HSV-2, and VZV, so that they can no longer transport nor shield the viral genomes. Capsid disassembly did not require proteases but depended on the ability of MxB to hydrolyse GTP and to dimerize. As the large tegument protein pUL36 links the capsid vertices to the other tegument proteins (*Crump, 2018*; *Dai and Zhou, 2018*; *Diefenbach, 2015*), and as an increasing amount of associated tegument proteins protected capsids against MxB-mediated disassembly, we propose that MxB attacks the capsids at their vertices. Our data suggest that MxB can bind to and disassemble incoming as well as progeny capsids, and thereby

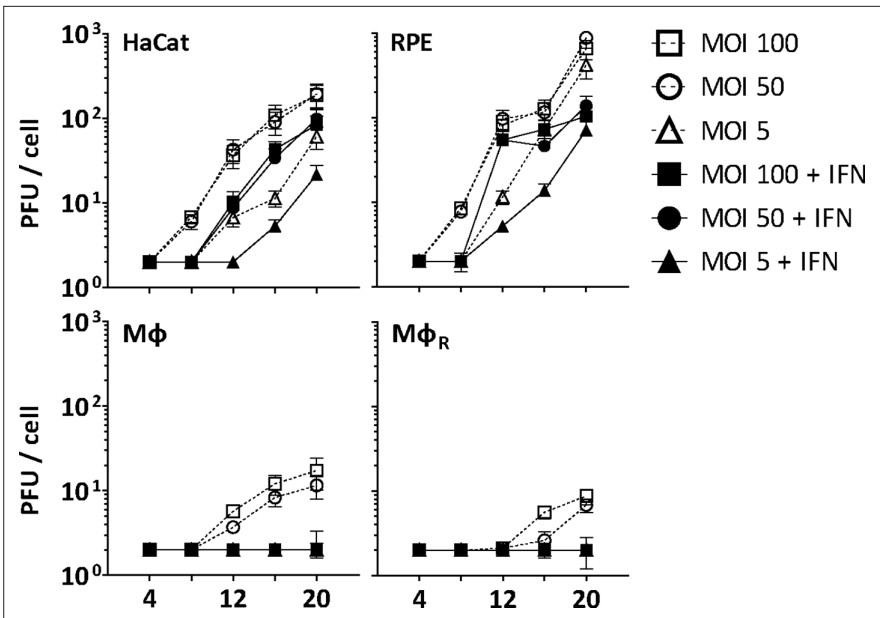

**Figure 1.** IFN restricts HSV-1 infection in keratinocytes, epithelial cells, and macrophages. HaCat, RPE, Mφ, or Mφ$_R$ cells were mock-treated or treated with human IFN-α (1000 U/mL) for 16 hr and were infected with HSV-1(17⁺) Lox at 2.5 × 106 (MOI 5), 2.5 × 107 (MOI 50), or 5 × 107 PFU/mL (MOI 100), and the amount of cell-associated and extracellular virions was titrated on Vero cells. Each data point represents the mean of the three technical replicates of the combined cell-associated and extracellular titers. The error bars represent the standard deviation.

might increase the sensing of cytosolic and nuclear viral genomes. Therefore, the MxB GTPase might be the sought-after capsid destroyer that acts upstream of cytosolic or nuclear sensors to promote viral genome detection and induction of innate immune responses.

## Results

### IFN induction prevents HSV-1 infection of macrophages

Before investigating capsid interactions with macrophage proteins, we compared HSV-1 infection in human keratinocytes (HaCat), pigment epithelial cells (RPE), and THP-1 cells at low, moderate, or high multiplicity of infection (MOI). We stimulated monocyte THP-1 cells with phorbol 12-myristate 13-acetate to differentiate them into a macrophage-like phenotype, and used them either directly (Mφ) or after a resting period of 3 days (Mφ$_R$). HSV-1 replicated productively in HaCat and RPE cells up to 20 hpi, while a pre-treatment with IFN delayed and reduced but did not prevent the production of infectious virions (*Figure 1*). Both Mφ and Mφ$_R$ released 10–100-fold less infectious HSV-1, and an IFN pre-treatment infection at all MOIs. Thus, Mφ and Mφ$_R$ restricted HSV-1 infection efficiently, and the induction of IFN-stimulated genes (ISGs) prevented any productive infection.

### IFN-induced protein changes in the cytosol of macrophages

To identify cytosolic macrophage proteins that might foster or restrict HSV-1 capsid functions, we prepared extracts from Mφ$_R$ or IFN-induced Mφ$_{IFN}$ to reconstitute capsid-host protein complexes as they might assemble in macrophages (*Figure 2—figure supplement 1*). Using subcellular fractionation and subsequent dialysis (*Figure 2—figure supplement 2A*), we depleted the extracts of nuclei and mitochondria (*Figure 2—figure supplement 2B*; pellet P1), cytoplasmic membranes such as Golgi apparatus, endoplasmic reticulum and plasma membrane (P1, P2), and small metabolites (S2, S3, S4). Furthermore, most of the cytoskeletal tubulin and actin sedimented into the first pellet (P1), while glyceraldehyde 3-phosphate dehydrogenase (GAPDH), a bona-fide cytosolic protein, remained soluble in the supernatants (S1, S2, S2', S3, S4). Next, we analyzed the proteomes of the Mφ$_R$ and IFN-induced Mφ$_{IFN}$ cytosols at low ATP/GTP concentration [ATP/GTP$^{low}$] by mass spectrometry (MS; *Supplementary file 1*). We detected 494 (*Figure 2—figure supplement 2C*; black circles)

of more than 600 reported IFN-inducible proteins (*Rusinova et al., 2013*). Of those, a Fisher's exact test identified the interferomeDB, and in particular GALM, COL1A1, LGALS3BP, NT5C3A, IFI44, IFIT2, IFIT3, GBP4, SRP9, IFIT5, DSP, and L3HYPDH as enriched by at least 2.8-fold (log$_2$ 1.5) in the Mφ$_{IFN}$ cytosol (*Figure 2—figure supplement 2C*; red). These changes might reflect IFN-induced transcriptional or translational regulation, post-translational modification, subcellular localization, or susceptibility to proteolysis, and show that the IFN induction had changed the cytosol proteome of the Mφ$_{IFN}$.

## HSV-1 capsids interact with specific cytosolic macrophage proteins

To search for cytosolic Mφ proteins whose interactions with HSV-1 capsids depend on their surface composition, we generated tegumented viral V$_{0.1}$, V$_{0.5}$, and V$_1$ capsids as well as D capsids with a reduced tegumentation (*Figure 2—figure supplement 1*). For this, we lysed extracellular particles released from HSV-1 infected cells with non-ionic detergent to solubilize the envelope proteins and lipids, and in the presence of 0.1, 0.5, or 1 M KCl to modify intra-tegument protein-protein interactions (*Anderson et al., 2014*; *Ojala et al., 2000*; *Radtke et al., 2010*; *Radtke et al., 2014*; *Wolfstein et al., 2006*, *Zhang and McKnight, 1993*). Furthermore, we dissociated tegument from V$_{0.1}$ capsids by a limited trypsin digestion to generate so-called D capsids. We then incubated similar amounts of different capsid types as calibrated by immunoblot for the major capsid protein VP5 (*Figure 2—figure supplement 2D*) with cytosol at ATP/GTP$^{low}$ from Mφ$_R$ or IFN-induced Mφ$_{IFN}$ for 1 hr at 37 °C. The capsid-host protein complexes assembled in vitro were harvested by sedimentation, and their interactomes were determined by quantitative MS (*Figure 2—figure supplement 1*). As before (*Radtke et al., 2010*; *Snijder et al., 2017*), the protein intensities were normalized across samples to the abundance of the major capsid protein VP5 (*Supplementary file 2*, host; *Supplementary file 3*, viral).

Of 2,983 proteins identified (*Supplementary file 2*), we detected 1816 in at least three of the four replicates in any of the eight different capsid-host protein complexes. Of those, 598 host proteins bound differentially to one capsid type over another (*Supplementary file 2*; fold change ≥2 (log$_2$ 1); permutation-based FDR ≤ 0.05). The HSV-1 capsids had recruited specifically 279 proteins of Mφ$_R$ and 390 of Mφ$_{IFN}$ cytosol of which 71 were shared. Hierarchical clustering analyses of the associated Mφ$_R$ or Mφ$_{IFN}$ proteins identified four major classes; for example one enriched on V over D capsids (*Figure 2—figure supplements 3–4*, top green) and one enriched on D over V capsids (*Figure 2—figure supplements 3–4*, bottom violet). Therefore, we further compared the capsid-host interactions of D capsids directly to V$_{0.1}$ (*Figure 2A, D*), V$_{0.5}$ (*Figure 2B and E*), or V$_1$ (*Figure 2C and F*) capsids, and identified 82 proteins of Mφ$_R$ (*Figure 2A, B and C*) and 141 of Mφ$_{IFN}$ (*Figure 2D, E and F*) with 35 being shared (*Supplementary file 2*; difference ≥2.83 fold (log$_2$ 1.5); FDR ≤ 0.01). The Mφ$_R$ capsid-host complexes included 12 and the ones of Mφ$_{IFN}$ 19 proteins listed in the interferome database (*Rusinova et al., 2013*; red in *Figure 2*). Gene ontology and pathway enrichment analyses showed that the identified 82 Mφ$_R$ (*Figure 3—figure supplement 1*) and 141 Mφ$_{IFN}$ (*Figure 3*) proteins included many proteins implicated in innate immunity, intracellular transport, nucleotide and protein metabolism, as well as intracellular signaling. Overall, the host proteomes of V$_{0.1}$ (red) and D (gray) capsids were rather distinct, but more similar for V$_{0.5}$ (blue) and V$_1$ (green) capsids (*Figure 3—figure supplement 1*, *Figure 3*). For example, V$_{0.1}$ capsids had recruited specifically the innate immunity proteins PIGR, IGHA1, BPIFA1 and DEFA3, but D capsids LRRFIP1, UFC, C3 and DCD from Mφ$_R$ cytosol. In Mφ$_{IFN}$, the D capsids were enriched for C3, C6, IGBP1, UBA5, UBXN1, UBE3A, and RNF123. These data suggest that protein domains displayed on different capsids interacted with specific cytosolic Mφ$_R$ or Mφ$_{IFN}$ proteins.

In these assays, the capsids interacted with several proteins already validated to promote or restrict HSV or VZV infection. Examples are the ESCRT-III co-factor VPS4 (*Cabrera et al., 2019*; *Crump et al., 2007*), EIF4H (*Page and Read, 2010*), the Kif2a subunit of kinesin-13 (*Turan et al., 2019*), the POLR1C subunit of RNA polymerase III (*Carter-Timofte et al., 2018*), the DNA protein kinase PRKDC (*Justice et al., 2021*), and DDX1 (*Zhang et al., 2011*). Moreover, the deubiquitinase USP7 (*Rodríguez et al., 2020*) and the ubiquitin ligases RNF123, TRIM72, UFC1 and UBE3A as well as the proteasome might regulate capsid functionality (*Huffmaster et al., 2015*; *Schneider et al., 2021*) or their degradation (*Horan et al., 2013*; *Sun et al., 2019*). These data show that HSV-1 capsids exposing a different tegument composition recruited specific cytosolic proteins from resting or IFN-induced macrophages.

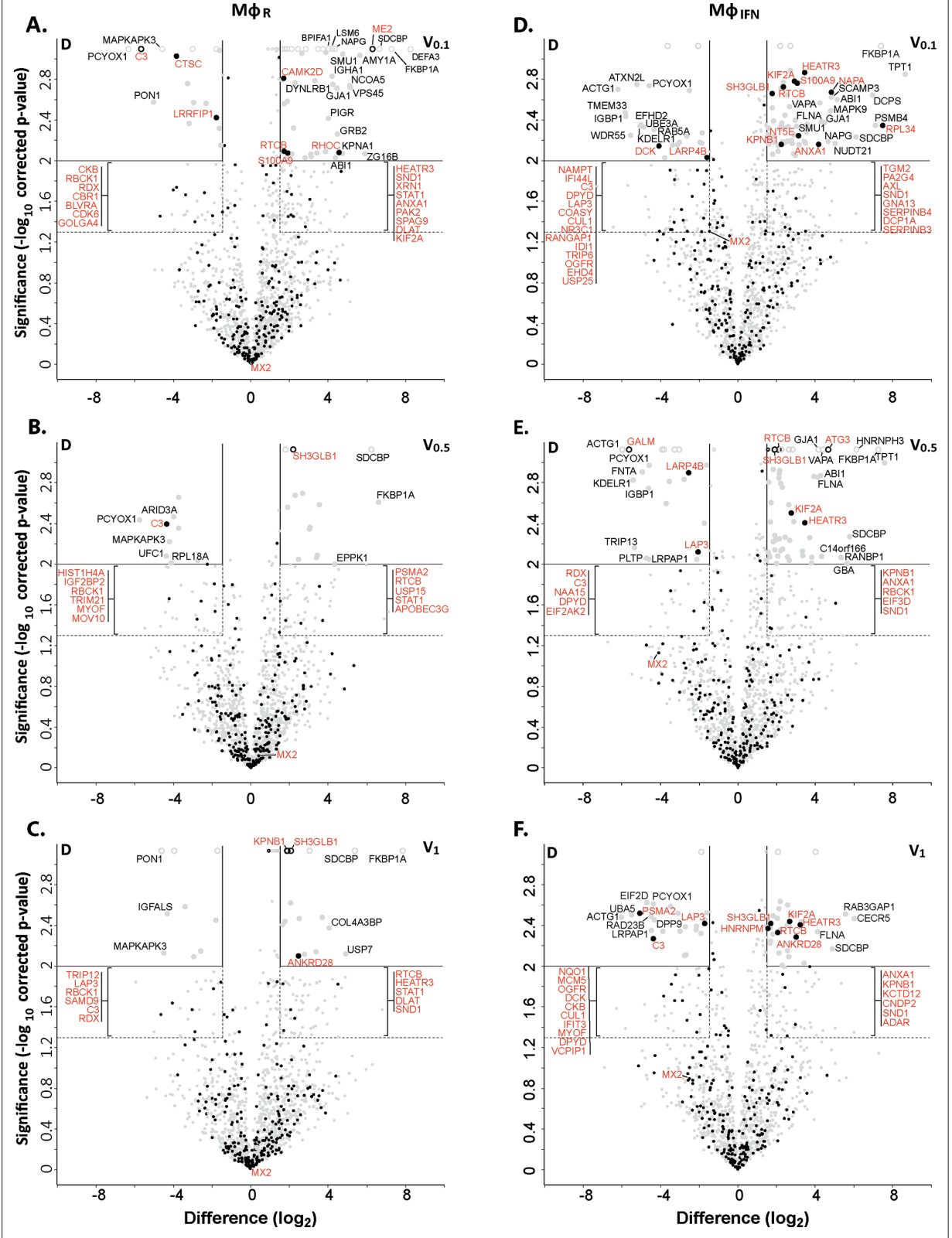

**Figure 2.** Cytosolic IFN-induced macrophage proteins binding to HSV-1 capsids. Volcano plots of iBAQs counts of proteins identified in capsid-host protein complexes assembled in cytosol from resting THP-1 φ cells (A - C) or treated with interferon-α (D - F) using $V_{0.1}$ (**A, D**), $V_{0.5}$ (**B, E**), or $V_1$ (**C, F**) capsids in comparison to D capsids. Proteins identified as highly specific interactions are indicated with larger symbols (log₂ difference ≥1.5; Welch's t-test, two-tailed, permutation-based FDR ≤ 0.01); those with a log₂ difference ≥4 are annotated. ISGs (interferome.org) are indicated by filled black circles,

*Figure 2 continued on next page*

*Figure 2 continued*

and are annotated in red if significantly enriched (permutation-based FDR ≤ 0.05, and $\log_2$ difference ≥1.5). Proteins with a q-value = 0 were imputed to - $\log_{10}$ q-value = 3.1 (maximum of the graph), and were indicated with empty circles.

The online version of this article includes the following source data and figure supplement(s) for figure 2:

**Figure supplement 1.** Experimental strategy to generate host protein-capsid complexes.

**Figure supplement 1—source data 1.** Characterization of macrophage subcellular fractionation.

**Figure supplement 2.** Characterization of cytosolic extracts and calibration of capsids.

**Figure supplement 3.** HSV-1 capsids interactomes.

**Figure supplement 4.** HSV-1 capsids interactomes.

## HSV-1 capsids recruit specific proteins responding to or regulating type I IFN

We next analyzed the $M\varphi_{IFN}$ samples in detail as IFN induction had prevented HSV-1 infection completely. We generated cluster maps for the 32 capsid-associated proteins belonging to the GO clusters *Response to type I IFN* or *Regulation of type I IFN production* (*Supplementary file 2*). V capsids recruited DHX9, HSPD1, and FLOT1 as well as proteins involved in the DNA damage response like PRKDC/DNA-PK, XRCC5, and XCCR6 from both, $M\varphi_R$ and $M\varphi_{IFN}$ cytosol (*Figure 4*). Interestingly, V capsids bound specifically to STAT1 in $M\varphi_R$, but to ADAR and IFIT2 in $M\varphi_{IFN}$ cytosol. D capsids were enriched for IFI16, OAS2, POLR1C, STAT2, and MxB (gene Mx2) in $M\varphi_{IFN}$ but not in $M\varphi_R$ (*Figure 4*, *Figure 4—figure supplement 1*). Particularly interesting was the discovery of MxB in these capsid-host protein complexes. MxB was significantly enriched on HSV-1 D capsids in $M\varphi_{IFN}$ cytosol, and the IFN treatment had the strongest impact on the interaction of MxB with capsids. Moreover, the calculated enrichment score for MxB on capsids was very high, although the MxB levels in the input cytosol were below the detection limit (undetected, *Figure 2—figure supplement 2*, *Supplementary file 1*). MxB but not its homolog MxA restricts infections of the herpesviruses HSV-1, HSV-2, MCMV, KSHV, and MHV-68, but its mode of action has not been elucidated (*Crameri et al., 2018*, *Liu et al., 2012*, *Schilling et al., 2018*, *Jaguva Vasudevan et al., 2018*). For these reasons, we investigated the interaction of human MxB with HSV-1 capsids further.

## MxB binds to capsids

We first characterized the MxB fractionation behavior during the cytosol preparation (*Figure 2—figure supplement 2*). As reported (*Goujon et al., 2013*; *Melén et al., 1996*), MxB was upregulated in IFN-induced $M\varphi_{IFN}$. MxB sedimented with nuclei and mitochondria as reported before (*Cao et al., 2020*), and also with cytoplasmic membranes. Moreover, MxB can assemble into cytosolic filaments (*Alvarez et al., 2017*) which might have been sedimented on their own. Both, after the addition of ATP and GTP (ATP/GTP$^{high}$) or the hydrolase apyrase (*Pilla et al., 1996*; ATP/GTP$^{low}$), a significant fraction of MxB remained soluble in the cytosol.

Next, we confirmed by immunoblotting that MxB co-sedimented with HSV-1 capsids which had been incubated in cytosols from $M\varphi_R$ or $M\varphi_{IFN}$. In line with the MS results, MxB bound better to D than to $V_{0.1}$, $V_{0.5}$, or $V_1$ capsids (*Figure 5A*). We next probed authentic nuclear capsids, namely empty A, scaffold-filled B, or DNA-filled C capsids, as well as tegumented $V_1$, $V_{0.5}$, $V_{0.1}$ or D capsids with cytosol of A549-MxB(1-715) epithelial cells expressing MxB(1-715). Nuclear A and C as well as $V_1$ and D capsids recruited MxB efficiently, while B, $V_{0.1}$ and $V_{0.5}$ capsids bound less MxB (*Figure 5B*). MxB did not sediment by itself, and also did not associate with agarose beads used as another sedimentation control (*Figure 5A and B*). These data indicate that MxB binds to specific structural features on the capsid surface.

In cells, MxB-mediated restriction of herpesvirus replication depends on its N-terminal 25 amino acid residues (NTE), its GTPase activity, and its capacity to form dimers (*Crameri et al., 2018*; *Schilling et al., 2018*; *Jaguva Vasudevan et al., 2018*). We incubated capsids with cytosols containing MxA, MxB(1-715), MxB(26-715) (*Melén et al., 1996*; *Melén and Julkunen, 1997*), MxB(K131A) with reduced GTP binding, MxB(T151A) lacking the GTPase activity, or MxB(M574D) unable to dimerize (*Alvarez et al., 2017*; *Fribourgh et al., 2014*; *King et al., 2004*; *Schilling et al., 2018*). In contrast to MxA, MxB(1-175), MxB(26-715), and MxB(M574D) co-sedimented with capsids to a similar extent.

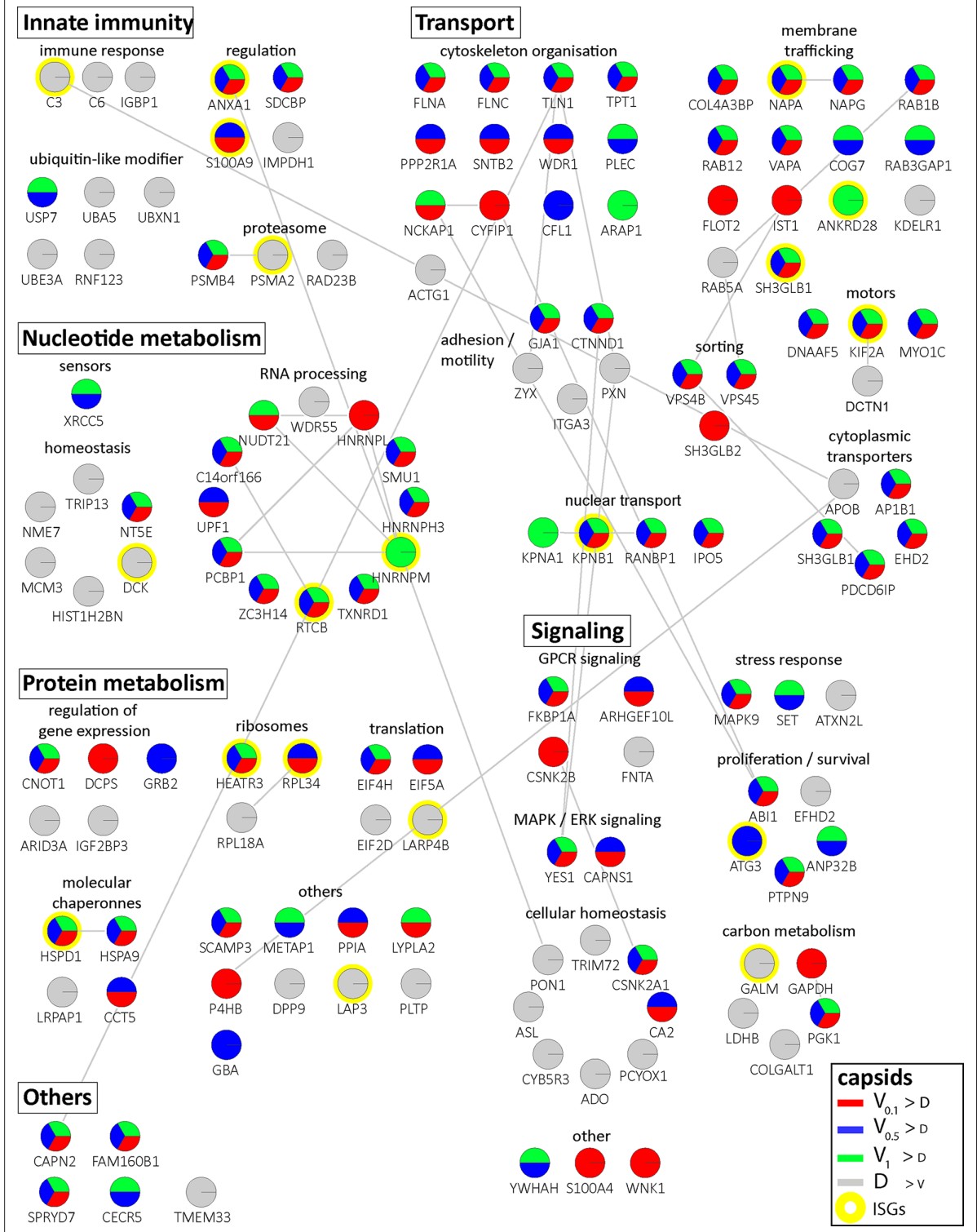

**Figure 3.** Cytosolic proteins of IFN-induced macrophages binding to HSV-1 capsids. Host proteins from cytosol of IFN-stimulated Mφ$_{IFN}$ (c.f. D, E, F; abundance log$_2$ difference larger than 1.5; significance permutation-based FDR smaller than 0.01) interacting with V$_{0.1}$, V$_{0.5}$, V$_1$, or D capsids were assembled into a functional interaction network of known protein-protein-interactions (gray lines; STRING database, confidence score of 0.7), and grouped according to their known functions (Gene Ontology, Pathway analysis). The Pie chart for each protein indicates its relative enrichment on V$_{0.1}$ (red), V$_{0.5}$ (blue), V$_1$ (green), or D capsids (gray).

The online version of this article includes the following figure supplement(s) for figure 3:

**Figure supplement 1.** Cytosolic proteins of resting macrophage binding to HSV-1 capsids.

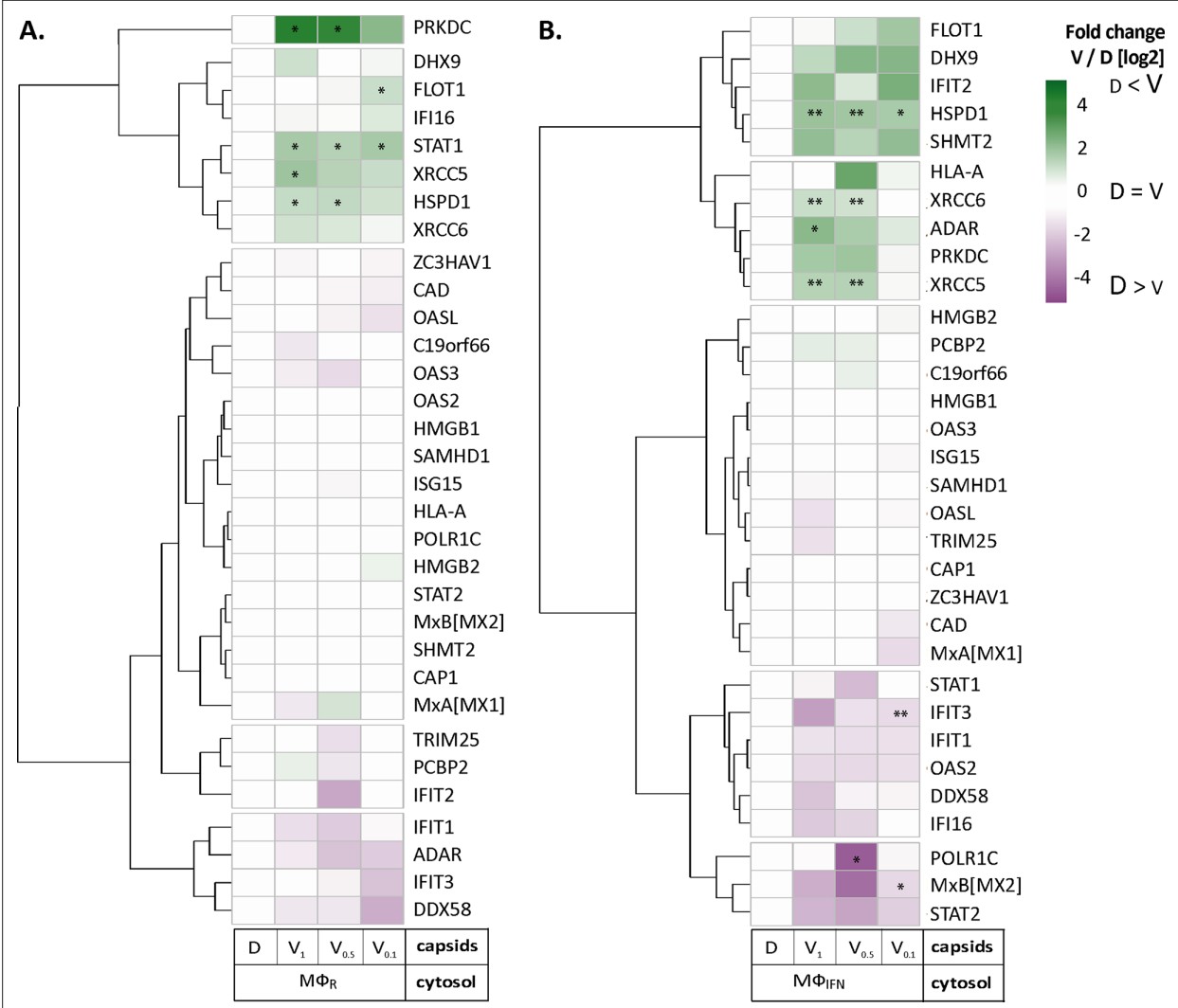

**Figure 4.** HSV-1 capsids associate with proteins involved in type I IFN response. Unbiased hierarchical clustered heat map showing the log₂ fold changes of IFN-induced proteins (GO type-I IFN) identified from capsids-host protein sediments from cytosol of resting Mφ, or IFN-induced Mφ_IFN macrophages. For each protein, the fold change was calculated based on their abundance (iBAQs) in $V_1$, $V_{0.5}$, and $V_{0.1}$ capsids as compared to their abundance in D capsids, using a linear scale from violet being the lowest to dark green being the highest. (*) and (**) design the proteins with an FDR corrected p-value ≤ 0.05 and ≤ 0.01, respectively.

The online version of this article includes the following figure supplement(s) for figure 4:

**Figure supplement 1.** HSV-1 capsids binds to a few ISG proteins.

Interestingly, MxB(K131A) did not bind to capsids, while MxB(T151A) bound even stronger (**Figure 5C**). These data suggest that conformational changes associated with GTP binding or hydrolysis contribute to MxB interaction with HSV-1 capsids.

## MxB disassembles capsids of alphaherpesviruses

Next, we tested whether MxB might affect HSV-1 capsid stability. While the previous capsid sedimentation assays were performed at ATP/GTP^low, they suggested that the GTP/GDP state of MxB might modulate its interaction with capsids. To test this experimentally, we supplemented the cytosols with 1 mM GTP, 1 mM ATP, and 7.5 mM creatine phosphate to maintain high ATP/GTP levels [ATP/GTP^high]. We resuspended sedimented capsid-host protein complexes and applied them onto EM grids (**Figure 2—figure supplement 1**), or we added isolated capsids directly onto EM grids and then placed them on a drop of cytosol to allow the formation of capsid-host protein complexes (**Figure 6A**). This direct *on-grid assay* required 50 times fewer capsids than the *sedimentation-resuspension assay*

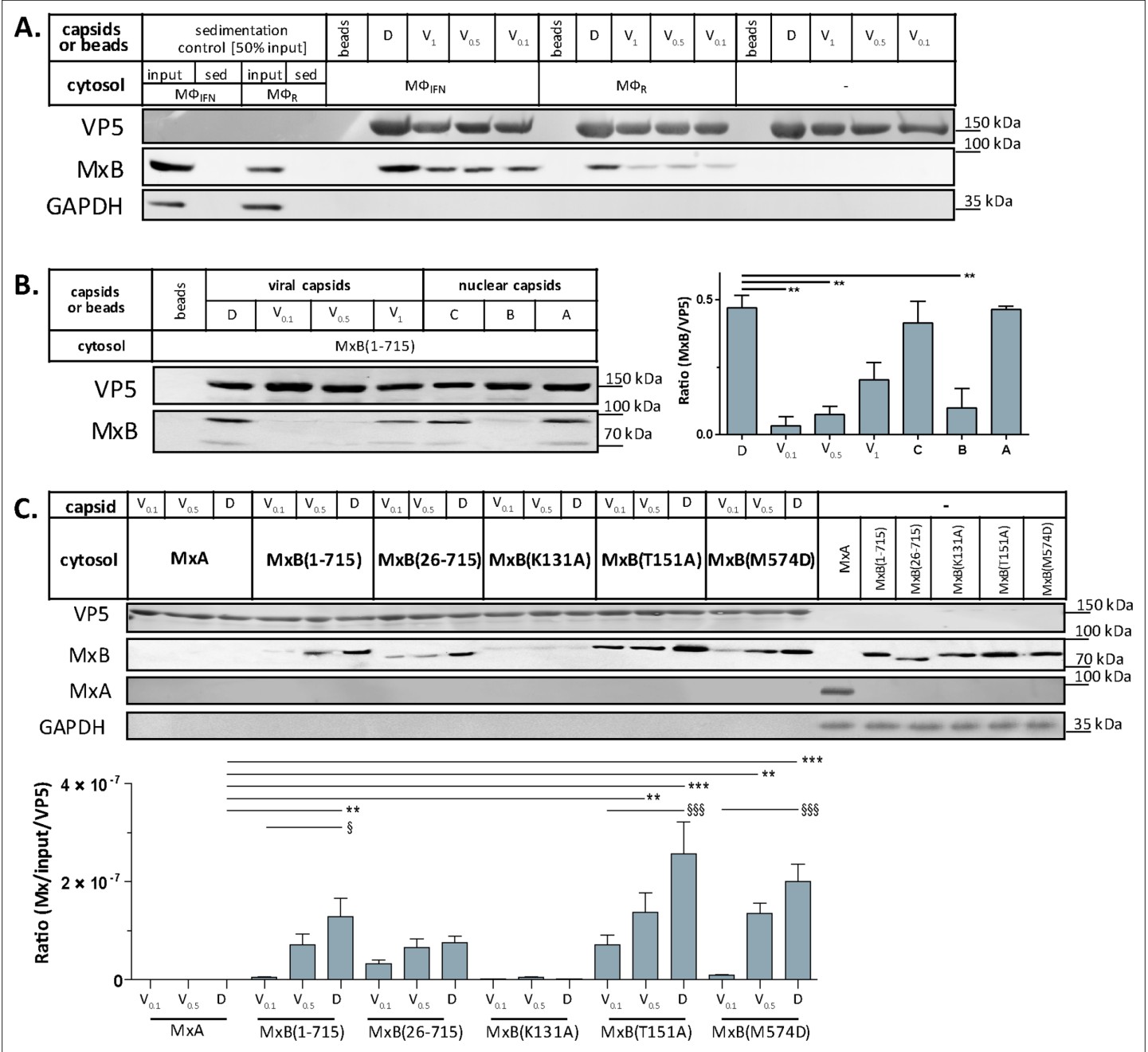

**Figure 5.** Tegumentation reduces MxB binding to HSV-1 capsids. The binding of MxB to viral $V_{0.1}$, $V_{0.5}$, $V_1$, or D, or to nuclear A, B, or C capsids was analyzed after incubation in 0.2 mg/mL cytosol prepared from (A; *Figure 5—source data 1*) THP-1 φ stimulated or not with IFN, or (B-C; *Figure 5—source data 1*; *Figure 5—source data 1*) A549 cells stably expressing MxA, MxB(1-715) full length, the short MxB(26-715), or MxB mutants defective in GTP-hydrolysis MxB(T151A), GTP-binding and hydrolysis MxB(K131A), or dimerization MxB(M574D). Sedimented capsid-host protein complexes were then analyzed by immunoblot for VP5 (capsid), MxB, MxA, and GAPDH as a loading control. As control cytosols were sedimented without capsids (A: sed), or with uncoated agarose beads (A, B: beads). The amounts of MxA/MxB found in the capsid-host protein complexes were quantified, and normalized to their respective VP5 levels. Error bars: SEM. summarized from three experiments. One asterisk denotes p < 0.05, two asterisks indicate p < 0.01 and three asterisks represent p < 0.001 as determined by Welch's t-tests comparisons.

The online version of this article includes the following source data for figure 5:

**Source data 1.** Cytosolic MxB from THP-1 cells co-sediments with capsids in *Figure 5A*.

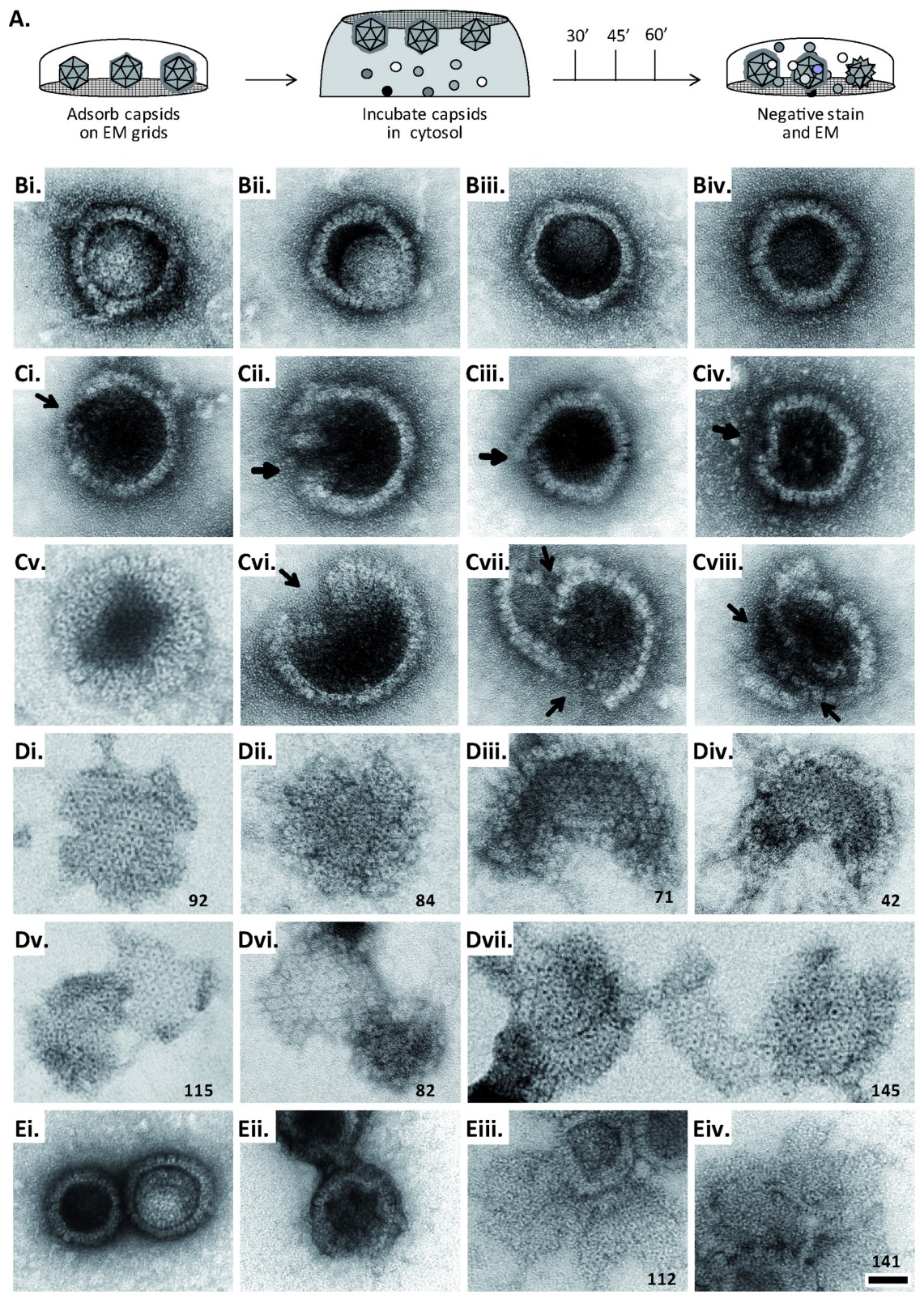

**Figure 6.** MxB induces disassembly of herpesviral capsids. (**A**) Experimental design: Capsids were adsorbed onto hydrophilic enhanced carbon-coated EM grids for 20 min at RT. The capsids were incubated in cytosol with ATP/GTP$^{high}$, and the incubation was stopped at different times by extensive washing. The samples were analyzed by EM after negative staining with uranyl acetate. (**B–D**) Capsids after incubation in cytosol derived from rested Mφ or IFN-induced Mφ$_{IFN}$ macrophages, or control or MxB(1-715) A549 expressing cells for 1 hr at 37 °C, and classified as (**B**) intact, (**C**) punched or

*Figure 6 continued on next page*

*Figure 6 continued*

(**D**) disassembled flat phenotypes. The number of capsomers per flat particle was counted, and is displayed at the bottom of each figures. (**E**) Nuclear VZV capsids remain intact (Ei) after incubation in the cytosol of A549 control cells, or but appear punched (Eii) or as flat shells (Eiii, Eiv) after incubation in the cytosol of A549 cells expressing MxB. Scale bar: 50 nm.

The online version of this article includes the following figure supplement(s) for figure 6:

**Figure supplement 1.** Capsid disassembly intermediates by anti-capsid immunoEM.

___

and allowed for time-course analyses. For both, we negatively contrasted the samples with uranyl acetate and analyzed them by electron microscopy.

When capsids were incubated with cytosol from A549 control cells not containing MxB, we saw mostly intact capsids with an appropriate diameter of about 125 nm, and an intact icosahedral morphology characterized by pentons at the vertices and hexons on the triangular capsid faces (*Figure 6B*). The capsids contained genomic DNA as the uranyl acetate used for negative contrast staining had not or only partially entered the capsid lumen. But a treatment with cytosol from IFN-induced $M\varphi_{IFN}$ or A549-MxB(1-715) cells dramatically impaired the capsid shell. Based on different MxB-induced morphological changes, we classified the capsid structures that we had identified by immunolabeling for capsid proteins (*Figure 6—figure supplement 1*) into three categories. *Intact capsids* (*Figure 6B*, Figure S6A) have an icosahedral morphology and include empty A, scaffold-filled B, and DNA-filled C capsids. *Punched capsids* are characterized by indentations on one or more vertices and an impaired icosahedral shape (*Figure 6C*, *Figure 6—figure supplement 1B*). *Flat shells* have completely lost their icosahedral shape (*Figure 6D*, *Figure 6—figure supplement 1C*). We estimated the number of capsomers on *flat shells* based on their area, and scored a structure with <100 capsomers as a half capsid and with ≥100 as one capsid (numbers in *Figure 6D*). Cytosols containing MxB(1-715) also disassembled capsids of HSV-2 (not shown) or VZV (*Figure 6E*) to *punched capsids* and *flat shells*. As MxB induced capsid disassembly of HSV-1, HSV-2, and VZV, these experiments suggest that MxB restricts the infection of herpesviruses by targeting their capsids.

## MxB requires GTP hydrolysis and dimerization to attack herpesviral capsids

Next, we further characterized the capsid disassembly activity of MxB by quantitative electron microscopy. Cytosol from IFN-induced $M\varphi_{IFN}$ disassembled more than 80% of the capsids within 1 hr while resting $M\varphi_R$ disassembled only about 40% (*Figure 7A*). Cytosol derived from A549 control cells had a minor effect on capsids, while cytosol from A549-MxB(1-715) cells disassembled capsids almost as efficiently as cytosol from $M\varphi_{IFN}$. Spiking cytosol from A549 control cells with an increasing percentage of A549-MxB(1-715) cytosol led to an increasing capsid disassembly with a majority of *punched capsids*, at 50% or 66% MxB cytosol, while incubation in pure A549-MxB(1-715) cytosol lead to more than 95% disassembly to mostly *flat shells* within 1 hr of incubation (*Figure 7B*). We then asked whether MxB had activated other host proteins to mediate capsid disassembly, or whether it was directly responsible. We prepared cytosol from A549-MxB(1-715)-MxB(26-715) expressing both untagged MxB proteins, or from A549-MxB-FLAG expressing both MxB(1-715)-FLAG and MxB(26-715)-FLAG. Both cytosols promoted capsid disassembly (MxB; MxB-FLAG in *Figure 7C*). An immunodepletion with anti-FLAG antibodies removed only the FLAG-tagged MxB proteins (*Figure 7—figure supplement 1*), and accordingly the disassembly activity from the A549-MxB-FLAG cytosol (MxB-FLAG FT), but not from the A549-MxB(1-715)-MxB(26-715) cytosol (MxB FT) containing both untagged MxB proteins.

We next tested at ATP/GTP$^{high}$ the effect of various MxB mutants on HSV-1 capsid stability. While full-length MxB(1-715) induced capsid disassembly, the MxB mutants impaired in GTPase activity (T151A), GTP binding (K131A), or dimerization (M574D) as well as cytosol with MxB at ATP/GTP$^{low}$ did not (*Figure 7D*). In contrast, the smaller MxB(26-715) protein lacking the NTE retained about 50% of the capsid disassembly activity. Furthermore, studying the stability of capsids pre-adsorbed *on-grid* in a time-course revealed a lag phase of about 30 min until broken capsids appeared with increasing rate (*Figure 7E*). The percentage of *punched capsids* reached a plateau at 50 min, while the amount of *flat shells* continued to increase (*Figure 7E*). Further experiments showed that MxB attacked D capsids more efficiently than tegumented $V_{0.5}$ capsids, of which about 70% resisted the MxB attack (*Figure 7F*). In contrast, the $V_{0.1}$ capsids seemed to be spared from MxB attack, since no

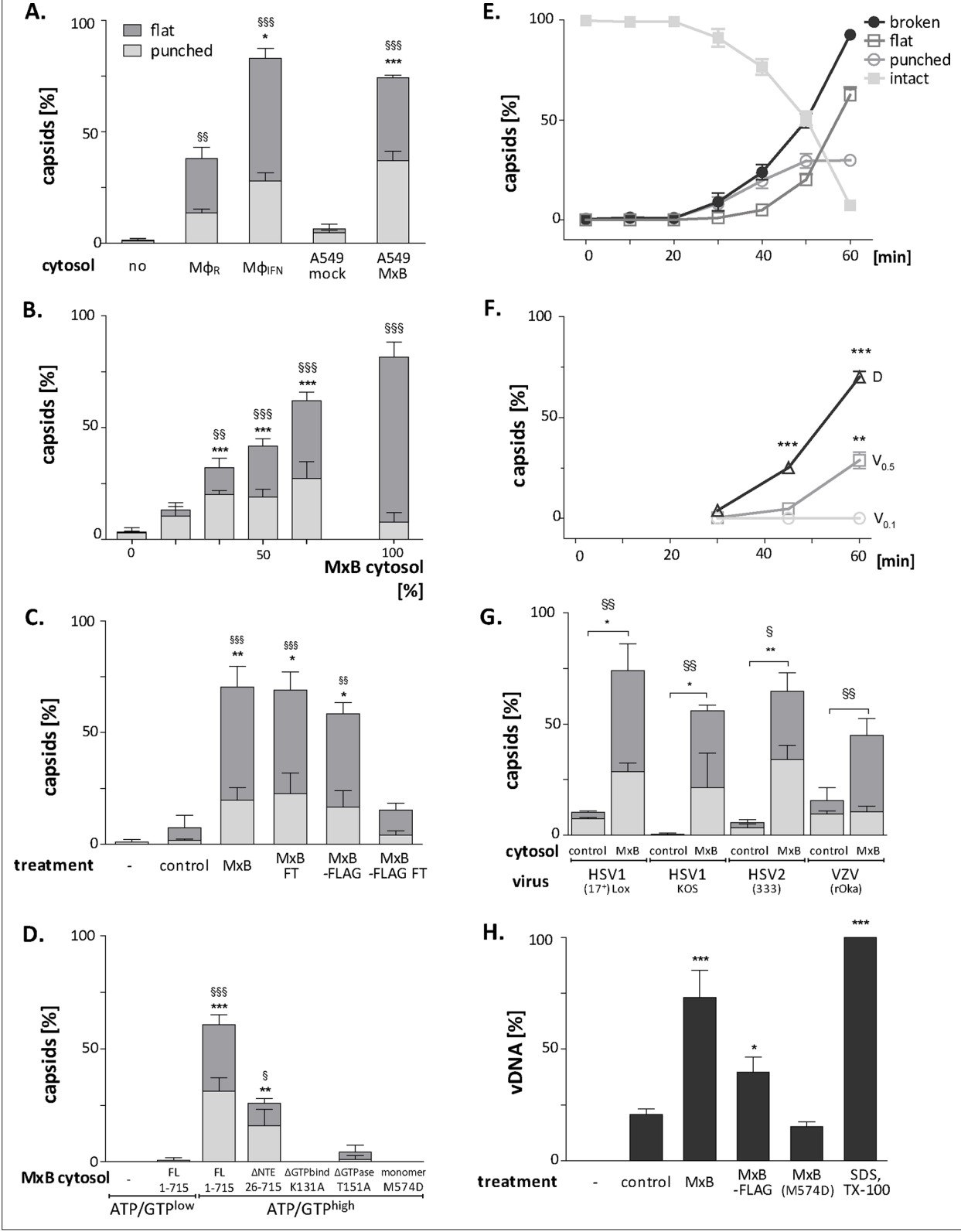

**Figure 7.** MxB GTP hydrolysis and dimerization required for capsid disassembly and vDNA release of viral genomes. HSV-1 (**A–H**), HSV-2 (**G**) or VZV capsids (**G**) were incubated with cytosol at ATP/GTP$^{high}$ for 1 hr or the indicated time (**E,F**) at 37 °C, and classified into *intact*, *punched* and *flat* capsids by electron microscopy (**A–G**), or the amount of released viral DNA was measured by qPCR (**H**). (**A**) Quantification of *punched* and *flat* D capsid shells in cytosol prepared from rested Mφ or IFN-induced Mφ$_{IFN}$ macrophages, or from control A549 (mock) or A549-MxB(1-715) cells. (**B**) Increasing amounts

*Figure 7 continued on next page*

*Figure 7 continued*

of MxB(1-715) [%] were added to control A549 cytosol, and the amounts of *punched* and *flat* capsids were quantified after incubation in these mixtures. (**C**) Cytosols of A549 cells expressing MxB(1-715) and Mx(25-715) or MxB(1-715)-FLAG and MxB(26-715)-FLAG were incubated with anti-FLAG antibodies coupled to magnetic beads, the flow-through fractions (FT) were harvested, capsids were treated with anti-FLAG treated or control cytosols, and the amount of punched and flat capsids were quantified. (**D**) Capsids were incubated in cytosols prepared from A549 cells expressing full-length (FL) MxB(1-715), MxB(26-715), MxB(K131A), MxB(T151A), or MxB(M574D) at ATP/GTP$^{low}$ or ATP/GTP$^{high}$ levels. (**E**) Time-course of MxB-induced disassembly of capsids pre-adsorbed onto EM grids, incubated with cytosol from A549-MxB(1-715). (**F**) Analysis of D, $V_{0.5}$, or $V_{0.1}$ capsids treated with MxB(1-175) cytosol for *broken* (*punched +flat*) capsids after negative stain and EM as described for panel E. (**G**) Quantification of MxB cytosol disassembly of D capsids of HSV-1(17$^+$)Lox, HSV-1(KOS), or HSV-2(333), or nuclear C capsids of VZV, after incubation in cytosol from A549-MxB(1-715) cells. (**H**) D capsids were incubated with different cytosols for 1 hr at 37 °C or treated with 1% SDS and 10% Tx-100 only, and the released DNA not protected by capsid shells was quantified by qPCR. Error bars: SEM from 100 capsids in three biological replicates. One symbol of *or § denotes p < 0.05, two p < 0.01, and three p < 0.001 as determined in One-way analysis of variance with a Bonferroni post-test, and comparing the relative amounts of (*) *punched* and (§) *flat* capsids, or indicating the differences with the mock-treated samples (*).

The online version of this article includes the following source data and figure supplement(s) for figure 7:

**Figure supplement 1.** Cytosol immunodepleted for MxB.

**Figure supplement 1—source data 1.** Western blot of MxB immunodepletion.

broken capsids appeared within an 1 hr treatment. Since MxB restricts infection of several herpesviruses (***Crameri et al., 2018***, ***Liu et al., 2012***, ***Schilling et al., 2018***, ***Jaguva Vasudevan et al., 2018***), we compared the impact of MxB on D capsids from HSV-1(17$^+$)Lox, HSV-1(KOS), HSV-2(333), or on nuclear C capsids from VZV(rOka). Capsids of these human alphaherpesviruses were all susceptible to MxB attack (***Figure 7G***).

## MxB attack leads to the release of viral genomes from capsids

Next, we determined how well the capsid shells protected the viral genomes against a DNA nuclease digestion. Capsids released three or two times more viral genomes in cytosols from MxB(1-715) or MxB-FLAG than from control or MxB(M574D) cells (***Figure 7H***). Together, these data indicate that the MxB GTPase disassembles the capsid shells and induces a release of viral DNA of several herpesviruses. Our experiments suggest that GTP binding and hydrolysis as well as dimerization contribute to MxB-mediated disassembly of alphaherpesvirus capsids. Its slow start with a lag of about 30 min indicates that the capsid attack might require some nucleating or cooperative reaction to assemble active MxB oligomers or an MxB-containing complex onto capsids.

### Tegument proteins protect against MxB attack

As complete tegumentation shielded $V_{0.1}$ capsids against destruction, while MxB bound to surface features exposed on $V_{0.5}$, A, C and D capsids, we compared the proteomes of the $V_{0.1}$, $V_{0.5}$, $V_1$, and D capsids. We calibrated the relative abundances of the 58 HSV-1 proteins detected to the normalized amounts of the major capsid protein VP5. The tegument compositions of $V_{0.1}$, $V_{0.5}$, and $V_1$ capsids were similar to each other but different from D capsids (***Figure 8***). The bona-fide capsid proteins VP21, VP24, VP22a, VP19c, and VP23 varied little among all capsid types. However, D capsids contain a bit less capsid surface proteins; namely VP26, the capsid specific vertex components (CSVC) pUL17 and pUL25, and to some extent the portal pUL6, and less of the major tegument proteins VP22, VP13/14, VP16, VP11/12 as well as other tegument proteins with ICP0, pUL36, and pUL37 being most susceptible to the trypsin treatment. Overall, there were little differences in the relative tegument protein amounts among $V_{0.5}$ and $V_1$ capsids. In contrast, $V_{0.1}$ capsids contained more tegument proteins, for example VP13/14, pUS3, and pUL16. All capsid preparations contained traces of membrane proteins and nuclear HSV-1 proteins contributing to DNA replication and packaging (***Figure 8—figure supplement 1***). These data further validated that a treatment with 0.5 or 1 M KCl during the detergent lysis of virions destabilized intra-tegument interactions. Furthermore, the limited trypsin digestion had reduced the capsid proteome further and increased the susceptibility to MxB attack.

## Discussion

Cell-type-specific defence mechanisms shape the arms race between proteins restricting or promoting nuclear targeting of incoming viral capsids and viral genome release into the nucleoplasm. We have

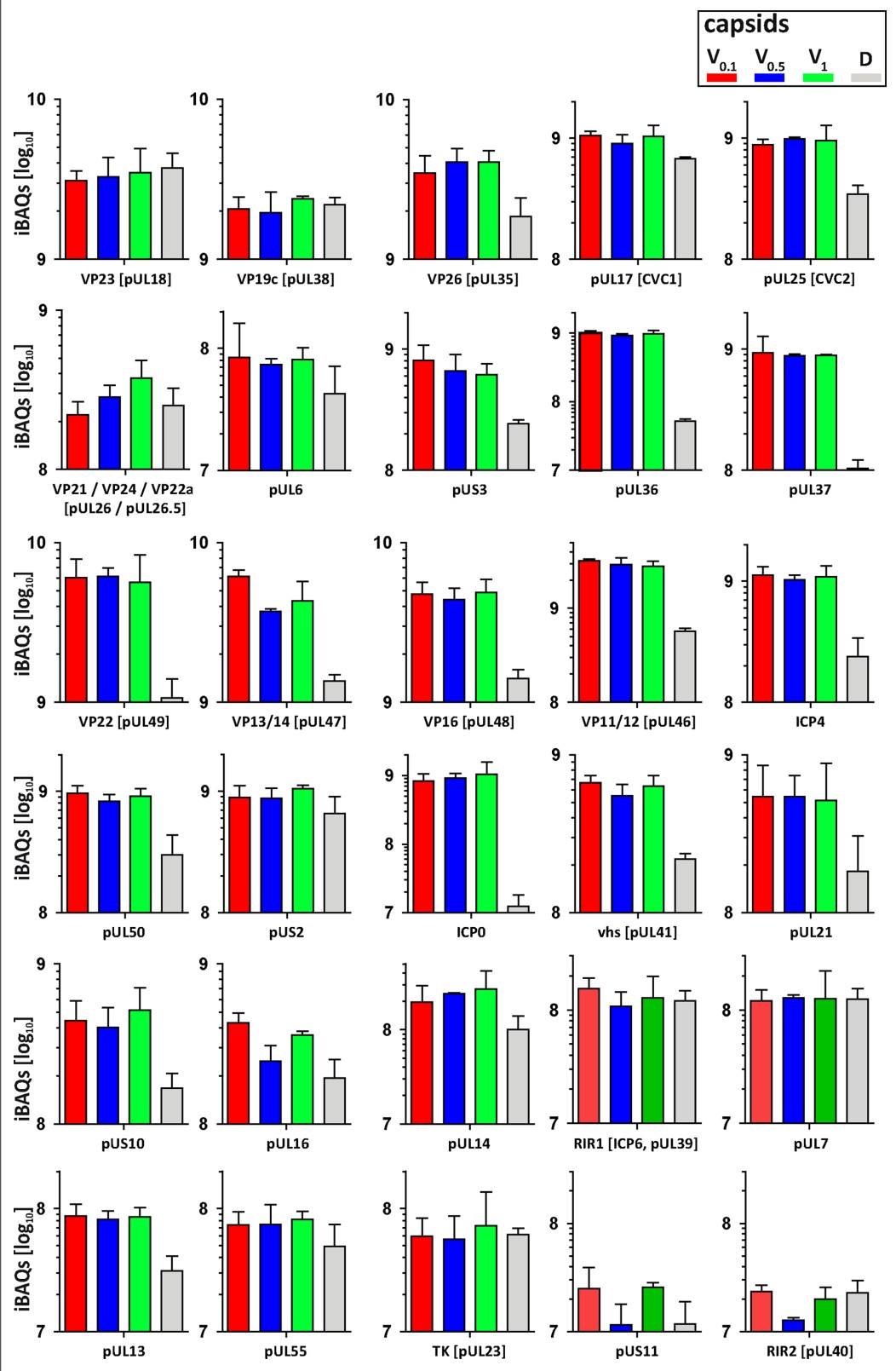

**Figure 8.** Structural and tegument characterization of $V_{0.1}$, $V_{0.5}$, $V_1$, and D capsids. The composition of HSV-1(17+) Lox derived $V_{0.1}$ (red), $V_{0.5}$ (blue), $V_1$ (green), and D (gray) capsids was analyzed by quantitative mass spectrometry in four biological replica. The sum of all the peptides intensities (iBAQ, intensity-based absolute quantification)

*Figure 8 continued on next page*

*Figure 8 continued*

of each viral protein known to participate in the structure of the capsids was normalized to the one of VP5 and displayed in a bar plot for each viral protein.

The online version of this article includes the following figure supplement(s) for figure 8:

**Figure supplement 1.** Membrane and non-structural proteins on V capsids versus D capsids.

developed biochemical assays to investigate functional interactions of viral capsids with host cell structures (*Radtke et al., 2014*), and analyzed here HSV-1 capsid-host protein complexes assembled in cytosols from resting $M\varphi_R$ or IFN-induced $M\varphi_{IFN}$ cells. We show that the IFN-inducible MxB GTPase bound to alphaherpesviral capsids, most likely to structural features around the capsid vertices, and disassembled herpesvirus capsids in a GTP-dependent fashion, and so that they no longer shielded the viral genomes. Capsid disassembly by MxB could reduce nuclear targeting of incoming capsids and genomes, but stimulate the activation of cytosolic DNA sensors and innate immune responses.

## Cytosolic IFN-induced macrophage proteins binding to HSV-1 capsids

IFN induction prevented HSV-1 infection of Mφ, and increased the cytosolic abundance of at least 12 proteins listed in the interferome database (*Rusinova et al., 2013*). Here, we assembled host protein-capsid complexes from HSV-1 capsids and cytosols of Mφ or $M\varphi_{IFN}$ cells as they might also form in cells. While our MS analyses showed that $V_{0.5}$ and $V_1$ capsids recruited unique but also common proteins, the proteomes of $V_{0.1}$ and D capsids were more distinct. These specific interactions are consistent with the notion that a treatment with 0.5 or 1 M KCl during the detergent lysis of virions destabilized intra-tegument interactions, that influenced, for example, the recruitment of dynactin, kinesin-1, and kinesin-2 from brain cytosol (*Ojala et al., 2000*; *Radtke et al., 2010*; *Wolfstein et al., 2006*). Moreover, these results are consistent with immunoelectron microscopy data showing that the surface of distinct V capsid types display different tegument epitopes (*Radtke et al., 2010*), and with cryoelectron tomography data revealing diminishing tegument densities from $V_{0.1}$, $V_{0.5}$, $V_1$, capsids to C capsids (*Anderson et al., 2014*). Thus, the surface features of $V_{0.1}$, $V_{0.5}$, and $V_1$ capsids differ as indicated by cryoelectron tomography, binding of anti-tegument antibodies, and the recruitment of distinct sets of cytosolic proteins from brain tissue (*Radtke et al., 2010*), or macrophages as shown here. Host proteins may bind to viral proteins in both states, when they are soluble in the cytosol or the nucleoplasm, or when they are associated with capsids. From host proteins shown here to bind to capsids, direct interactions with tegument proteins have already been reported; for example USP7 binding to ICP0 (*Everett et al., 1997*) or EIF4H binding to vhs (pUL41; *Page and Read, 2010*). Furthermore, proteins involved in intracellular trafficking or virus assembly associated particularly with tegumented V capsids. For example, importin α5 (*KPNA1*) might mediate capsid targeting to the nuclear pores (*Döhner et al., 2018*; *Döhner et al., 2021*), while RAB1B contributes to the envelopment of cytosolic HSV-1 capsids (*Zenner et al., 2011*).

## MxB binding to alphaherpesviral capsids

In addition to MxB, the host-capsid complexes included other antiviral proteins which in turn might be counteracted by HSV-1 proteins. Several $M\varphi_{IFN}$ proteins already know to restrict herpesviruses, for example STAT2, POLR1C, IFI16, DDX58 (RIG-I), and OAS2 (*Kurt-Jones et al., 2017*; *Lum and Cristea, 2021*; *Ma et al., 2018*), bound preferentially to D capsids. As it was not known how MxB might restrict herpesviral infection (*Crameri et al., 2018*; *Schilling et al., 2018*; *Jaguva Vasudevan et al., 2018*), we investigated its association with capsids further. B capsids are less sturdy and have not undergone the structural changes that stabilize the A and C capsids (*Roos et al., 2009*; *Sae-Ueng et al., 2014*; *Snijder et al., 2017*). Intriguingly, this stabilization depends on the CSVC proteins pUL17 and pUL25 (*Sae-Ueng et al., 2014*; *Snijder et al., 2017*), which are present on B, A, and C capsids (*Anderson et al., 2014*; *Radtke et al., 2010*; *Snijder et al., 2017*). As MxB bound to A, C and D, but not to B capsids, it might recognize surface features formed during capsid stabilization, e.g. matured CSVCs or portals, which are increasingly shielded on tegumented $V_1$, $V_{0.5}$, and $V_{0.1}$ capsids.

MxA and MxB GTPases inhibit several viruses by blocking early steps of infection (*Haller et al., 2015*). MxB binding to HIV capsids depends on its N-terminal region (NTR) of about 90 residues and the GTPase domain (*Betancor et al., 2019*; *Fricke et al., 2014*; *Smaga et al., 2019*; *Xie et al., 2021*).

Similarly, HSV-1 capsids bound MxB(1-715) and to a lesser extent MxB(26-715). But in contrast to HIV capsids (*Betancor et al., 2019*; *Xie et al., 2021*), HSV-1 capsids recruited also the GTPase deficient MxB(T151A) and the monomeric MxB(M574D). These data indicate that the interaction of MxB with HSV-1 capsids depends on the NTE of 25 residues, its GTP/GDP status, but not on its dimerization.

## MxB induced disassembly of alphaherpesviral capsids

HSV-1 capsid disassembly did not require proteolysis as the cytosols contained protease inhibitors, but may be modulated by other host proteins as there was a considerable lag phase. MxB did not attack fully tegumented $V_{0.1}$ capsids, while $V_{0.5}$ or D capsids were more susceptible. The large tegument protein pUL36 links other tegument proteins to the capsids; it is tightly associated with pUL17 and pUL25 at the CSVCs at the pentons, and it extends toward the twofold symmetry axes connecting neighboring capsid faces (*Coller et al., 2007*, *Liu et al., 2017*, *Newcomb and Brown, 1991*, *Roos et al., 2009*, *Schipke et al., 2012*). Our electron microscopy data suggest that MxB attacked the fivefold symmetry axes as the *punched capsids* had dramatic dents on the capsid vertices. MxB might furthermore attack the portal cap, a cap of HSV1-pUL25 or its homologs in other herpesviruses, which seals the pUL6 portal after DNA packaging is completed (*Liu et al., 2019*, *McElwee et al., 2018*; *Naniima et al., 2021*). The high internal capsid pressure due to the negatively charged genome (*Bauer et al., 2013*; *Roos et al., 2009*) could support the MxB attack from the outside. The limited trypsin treatment might have primed the D capsids for disassembly, as they contained less pUL36, pUL17, pUL25, and pUL6 than the V capsids. However, MxB also attacked $V_{0.5}$ capsids that resemble cytosolic capsids during nuclear targeting or after nuclear egress (*Ojala et al., 2000*; *Wolfstein et al., 2006*; *Radtke et al., 2010*; *Anderson et al., 2014*); just not as fast, and not as efficient. Altogether, these results suggest that increasing tegumentation protects incoming and newly assembled capsids, possibly by masking the MxB target structure, or by inhibiting its GTPase cycle.

The MxB-mediated capsid disassembly required its NTE(1-25), GTP hydrolysis, and dimerization. For the homologous MxA GTPase that limits infection of many RNA viruses (*Haller et al., 2015*), *Gao et al., 2011* proposed a restriction mechanism that involves GTP hydrolysis and a mechano-chemical coupling within ring-like oligomers with the GTPase domains being exposed on their outer diameter (*Gao et al., 2011*). Similarly, MxB can also assemble into helical tubes with the NTE and the GTPase domain oriented outwards (*Alvarez et al., 2017*). Accordingly, MxB monomers and dimers might associate with the capsid vertices and insert between the hexons of neighboring capsid faces. A further oligomerization of MxB and/or conformational changes associated with GTP hydrolysis might then exert destabilizing forces onto the capsid shells, and ultimately push the capsid faces apart.

## Does MxB induce capsid disassembly in cells?

Future studies need to investigate whether MxB also induces the disassembly of herpesviral capsids in cells. Upon docking of an incoming capsid to a NPC, the pUL25 portal cap is supposed to be displaced, the pUL6 portal to be opened, and the DNA to be ejected from the capsid into the nucleoplasm due to this intramolecular repulsion (*Brandariz-Nuñez et al., 2019*; *Döhner et al., 2021*; *Ojala et al., 2000*; *Rode et al., 2011*). In uninfected cells, there is a low amount of constitutively expressed MxB localized at the NPCs (*Crameri et al., 2018*; *Kane et al., 2018*; *Melén and Julkunen, 1997*), which might dislodge the portal cap and open the capsid portal on the incoming capsid to release the incoming genome into the nucleoplasm.

Crameri et al., proposed that the higher amounts of IFN-induced MxB may block cytosolic capsid transport, genome uncoating at the NPCs, and/or the release of viral genomes into the nucleoplasm, which is consistent with our biochemical data demonstrating MxB binding to HSV-1 capsids (*Crameri et al., 2018*). MxB-mediated disassembly of capsids that we report here would further reduce capsid targeting to the NPCs and genome release into the nucleoplasm. Accordingly, there are fewer HSV-1 capsid puncta in MxB expressing cells (*Crameri et al., 2018*). Consistent with our data on capsid disassembly with MxB(26-715), MxB(K131A), or MxB(M574D), restricting the infection of HSV-1, MCMV, and MHV68 also requires the NTE, GTP hydrolysis, and dimerization of MxB (*Crameri et al., 2018*; *Schilling et al., 2018*). Restriction of HIV infection depends also on MxB NTE and on MxB dimerization, while the role of its GTPase function requires further investigation (*Buffone et al., 2015*; *Fricke et al., 2014*; *Goujon et al., 2014*; *Schulte et al., 2015*; *Xie et al., 2021*). It will be interesting to determine whether MxB only competes for the binding of

host factors required for HIV intracellular trafficking, such as microtubule motor adaptors (BICD2; FEZ-1) or nucleoporins (reviewed in *Temple et al., 2020*), or whether it also induces HIV capsid disassembly.

Our data together with Schilling et al., and Crameri et al., suggest that the IFN-inducible MxB restricts HSV-1, HSV-2, VZV, and possibly other herpesviruses, by promoting efficient capsid disassembly (*Schilling et al., 2018*; *Crameri et al., 2018*). We cannot exclude that a surplus of capsid- and NPC-associated MxB imposes further restrictions on intracellular transport and genome release into the nucleoplasm. However, if MxB(1-715) would disassemble viral capsids before they are oriented properly with their portal toward the NPCs, their genomes would end up in the cytosol and would not be delivered into the nucleoplasm. There are fewer incoming cytoplasmic capsids in cells expressing MxB (*Crameri et al., 2018*), and incoming VP5 is ubiquitinated and degraded by proteasomes in macrophages (*Horan et al., 2013*; *Sun et al., 2019*). Therefore, capsid disassembly intermediates might be degraded in cells, while we could characterize them in our biochemical cell-free assays in which proteases had been blocked.

The viral genomes exposed after MxB-induced capsid disassembly might be degraded by the DNase TREX1 (*Sun et al., 2019*), or stimulate the DNA sensors AIM2, cGAS, or IFI16, and the induction of antiviral host mechanisms. As an inoculation with destabilized HIV-1 capsids leads to an increased activation of the DNA sensor cGAS (*Sumner et al., 2020*), the IFN-induced increased MxB expression might lead to a similar outcome in cells infected with herpesviruses. Accordingly, MxB may not only restrict herpesviruses by capsid disassembly, but also increase the exposure of viral genomes to cytosolic DNA sensors, which in turn would induce an IFN response, inflammation as well as innate and adaptive immune responses. Thus, MxB could be the long sought-after capsid sensor that destroys the sturdy herpesvirus capsids, and possibly HIV cores and other viral capsids, to promote host viral genome sensing.

# Materials and methods
## Cells
All cells were maintained in a humidified incubator at 37 °C with 5% $CO_2$, passaged twice per week, and were tested negative for mycoplasma contamination. BHK-21 (ATCC CCL-10) and Vero cells (ATCC CCL-81) were cultured in MEM Eagle with 1% NEAA (Cytogen, Wetzlar, Germany) and 10% or 7.5% (v/v) FBS, respectively (Good Forte; PAN-Biotech, Aidenbach, Germany). HaCat (*Boukamp et al., 1988*; kind gift from Detlef Neumann, Hannover Medical School, Hannover, Germany) and hTERT RPE-1 (RPE; CRL-2302, Clontech) were cultured in DMEM Gibco (Invitrogen) with 7.5% or 10% (v/v) FBS, respectively (Capricorn Scientific, Ebsdorfergrund, Germany). THP-1 cells (ATCC TIB-202; kind gift from Walther Mothes, Yale University, New Haven, USA) were cultured in RPMI Medium 1640 (Thermo Fisher Scientific, Waltham, Massachusetts, United States) with 10% FBS (Thermo Fisher Scientific, Waltham, Massachusetts, United States). THP-1 were stimulated with 100 nM phorbol 12-myristate 13-acetate (PMA; Sigma-Aldrich, Germany) for 48 hr and used immediately (Mφ) or after 3 days of rest (Mφ$_R$). The cells were cultured with 1000 U/mL human type I IFN-α2a (Mφ$_{IFN}$; R&D Systems, Minneapolis, Minnesota, USA) or left untreated for 16 hr.

A549-derived cells (ATCC CCL-185) were cultured in DMEM with 10% FCS. In addition to A549 control cells, we used A549 cell lines stably expressing MxB(1-715), MxB(1–715/K131A), MxB(1–715/T151A), MxB(1–715/M574D), MxB(26-715), or MxA(1-662) upon transduction with the respective pLVX vectors with an engineered Kozak sequence to favor expression of the MxB(1-715) over the MxB(26-715) proteins (*Schilling et al., 2018*). Furthermore, we generated A549-MxBFLAG cells expressing MxB(1-715)FLAG and MxB(26-715)FLAG, both tagged with the FLAG epitope (GACT ACAAAGACGATGACGACAAG) at the C-terminus of MxB (GenBAnk NM_002463), and A549-MxB(1-715)-MxB(26-715) cells expressing untagged MxB(1-715) and MxB(26-715) using the pLKOD-Ires-Puro vector (Clontech Takara Bio, Mountain View, United States). MeWo cells (kind gift from Graham Ogg; University of Oxford, Oxford, UK) were cultured in MEM with 10% FCS, NEAA, and 1 mM sodium pyruvate. None of the cells used in this study were identified in the list of commonly misidentified cell lines (International Cell Line Authentication Committee; https://iclac.org).

## Viruses

Virus stocks of HSV-1(17⁺)Lox (*Sandbaumhüter et al., 2013*), HSV-1 strain KOS (*Warner et al., 1998*; kind gift from Pat Spear, Northwestern Medical School, Chicago, USA), and HSV-2 strain 333 (*Warner et al., 1998*; kind gift from Helena Browne, Cambridge University, Cambridge, UK) were prepared as reported before (*Döhner et al., 2006*, *Grosche et al., 2019*). Extracellular particles were harvested from the supernatant of BHK-21 cells infected with 3–4 x $10^4$ PFU/mL (MOI of 0.01 PFU/cell) for 2–3 days until the cells had detached from the culture flasks, and plaque-titrated on Vero cells. VZV rOka (kind gift from Jeffrey Cohen, NIH, Bethesda, US) was maintained in infected MeWo cells (*Cohen and Seidel, 1993*; *Hertzog et al., 2020*). After 2–4 days, the VZV-infected cells as indicated by cytopathic effects were harvested, mixed with naive MeWo cells at a ratio of 1:4 to 1:8 for continued culture. Aliquots of frozen infected cells were used to inoculate cultures used for capsid preparation.

## HSV-1 infection

THP-1 were seeded at $2.5 \times 10^5$ cells per six-well, treated with 100 nM PMA (Sigma-Aldrich, Germany) for 48 hr, and used immediately (Mφ) or after 3 days of rest (Mφ$_R$). The cells were then induced with 1000 U/mL of IFN-α (Mφ$_{IFN}$) or left untreated for 16 hr. On the next day, they were inoculated with HSV-1(17⁺)Lox at $2.5 \times 10^6$, $2.5 \times 10^7$, or $5 \times 10^7$ PFU/mL (MOI of 5, 50, or 100 respectively) in $CO_2$-independent medium (Gibco Life Technologies) supplemented with 0.1% (w/v) cell culture grade fatty-acid-free bovine serum albumin (BSA; PAA Laboratories GmbH) for 30 min, and then shifted to regular culture medium at 37 °C and 5% $CO_2$. At the indicated times, the cells and the corresponding media were harvested separately and snap-frozen in liquid nitrogen. These samples as well as and HSV-1 and HSV-2 inocula were titrated on Vero cells (*Döhner et al., 2006*, *Grosche et al., 2019*).

## Preparation of V$_{0.1}$, V$_{0.5}$, and V$_1$ and D capsids

Extracellular HSV-1 or HSV-2 particles were harvested by sedimentation at 12,000 rpm for 90 min at 4 °C (Type 19 rotor, Beckman-Coulter) from the medium of BHK-21 cells (40 × 175 cm² flasks; 2–2.5 x $10^7$ cells/flask) infected with 0.01 PFU/cell (2–6.7 x $10^4$ PFU/mL) for 2.5 days. The resulting medium pellets (MP) were resuspended in 2 mL of MKT buffer (20 mM MES, 30 mM Tris-HCl, 100 mM KCl, pH 7.4), treated with 0.5 mg/mL trypsin (Sigma-Aldrich, Germany) at 37 °C for 1 hr which was then inactivated with 5 mg/mL trypsin inhibitor from soybean (SBTI; Fluka, Switzerland) for 10 min on ice (*Ojala et al., 2000*; *Radtke et al., 2010*; *Radtke et al., 2014*; *Wolfstein et al., 2006*. These samples were then mixed with an equal volume of 2-fold lysis buffer (2% TX-100, 20 mM MES, 30 mM Tris, pH 7.4, 20 mM DTT, 1 x protease inhibitor cocktail [PIs, Roche cOmplete] with 0.2 M, 1 M or 2 M KCl; *Radtke et al., 2014*). The samples were layered on top of 20% (w/v) sucrose cushions in 20 mM MES, 30 mM Tris, pH 7.4 with 10 mM DTT, PIs with the respective KCl concentration, and sedimented at 110,000 g for 20 min at 4 °C (TLA-120.2 rotor, Beckman-Coulter). The supernatants and the cushions containing solubilized viral envelope and tegument proteins were carefully removed. The pellets were resuspended in BRB80 (80 mM PIPES, pH 6.8, 12 mM $MgCl_2$, 1 mM EGTA) with 10 mM DTT, PIs, 0.1 U/mL protease-free DNase I (Promega, USA), and 100 mg/mL protease-free RNase (Roth GmbH, Germany) for 1 hr at 37 °C and then overnight at 4 °C. The capsids were sedimented at 110,000 g for 15 min at 4 °C (TLA-120.2) and resuspended in capsid binding buffer (CBB: 5% [w/v] sucrose, 20 mM HEPES-KOH, pH 7.3, 80 mM K-acetate, 1 mM EGTA, 2 mM Mg-acetate, 10 mM DTT and PIs) by ultrasound tip sonication at 40 W for about 5 × 5 s on ice. Furthermore, we treated V$_{0.1}$ capsids for 40 min at 37 °C with 10 µg/mL trypsin in CBB lacking PIs to generate D capsids by limited digestion. After the addition of 5 mg/mL SBTI for 10 min on ice to block the trypsin activity, the D capsids were sedimented at 110,000 x g and 4 °C for 15 min (TLA-120.2), and resuspended in CBB with PIs.

## Preparation of nuclear A, B, and C capsids

HSV-1 nuclear capsids were prepared from 40 × 175 cm² flasks with BHK-21 cells infected with 0.01 PFU/cell (3–4 x $10^4$ PFU/mL) for about 2.5 days (*Anderson et al., 2014*; *Radtke et al., 2010*; *Radtke et al., 2014*; *Snijder et al., 2017*; *Wolfstein et al., 2006*). VZV nuclear capsids were harvested from infected MeWo cells cultured in 5–10 x 175 cm² flasks at maximum syncytia formation but before cell lysis. The cells were harvested, resuspended in MKT buffer (20 mM MES, 30 mM Tris, pH 7.4, 100 mM KCl), snap-frozen, and stored at –80 °C. Nuclear A, B, and C capsids were separated by sedimentation at 50,000 x g and 4 °C for 80 min (SW40Ti, Beckman Coulter) on linear 20% to 50% sucrose gradients

in TKE buffer 20 mM Tris, pH 7.5, 500 mM KCl, 1 mM EDTA; diluted in three volumes of TKE supplemented with 2 mM DTT and PIs (Roche cOmplete). The capsids were sedimented in BSA-coated centrifuge tubes at 110,000 g at 4 °C for 20 min (TLA-120.2), resuspended in BRB80 buffer supplemented with 100 mg/mL RNase (Roth, Germany), 0.1 U/mL DNase I (M6101, Promega, USA), 10 mM DTT, and PIs, sedimented again, and resuspended in CBB with PIs.

## Calibration of capsid concentration

To calibrate the amount of capsid equivalents ($CAP_{eq}$) among different experiments, we compared all capsid preparations used in this study with a calibration curve generated from the same starting preparation. The capsids were suspended in sample buffer (1% [w/v] SDS, 50 mM Tris-HCl, pH 6.8, 1% [v/v] β-mercaptoethanol, 5% [v/v] glycerol, PIs [Roche cOmplete]), and adsorbed to nitrocellulose membranes (BioTrace, Pall Laboratory) using a 48-slot suction device (Bio-DOT-SF, Bio-Rad, Hercules, California, USA). The membranes were probed with a polyclonal rabbit serum raised against purified HSV-1 nuclear capsids (SY4563; *Supplementary file 4*; *Döhner et al., 2018*) followed by secondary antibodies conjugated to fluorescent infrared dyes (donkey-anti-rabbit IgG-IRDye1 800CW; *Supplementary file 3*), and documented with an Infrared Imaging System (Odyssey, Image Studio Lite Quantification Software, LI-COR Biosciences, Lincoln, Nebraska, USA). MPs harvested from one 175 cm² flasks of BHK-21 cells infected with HSV-1 contained about $0.5–1 \times 10^9$ PFU/mL, and $0.75–1.5 \times 10^9$ $CAP_{eq}$/mL. A nuclear HSV-1 capsid fraction prepared from one 175 cm² flask contained about $0.5–1 \times 10^7$ $CAP_{eq}$ of A capsids, $1–2 \times 10^7$ $CAP_{eq}$ of B capsids, and $0.5–0.75 \times 10^7$ $CAP_{eq}$ of C capsids, and a nuclear VZV fraction from one 175 cm² flasks of MeWo cells $2–4 \times 10^5$ $CAP_{eq}$ of A capsids, $0.5–1 \times 10^6$ $CAP_{eq}$ of B capsids, and $0.8–1.6 \times 10^7$ $CAP_{eq}$ of C capsids. Capsid-host protein complexes were assembled *in-solution* using $7.5 \times 10^8$ $CAP_{eq}$/condition for MS and immunoblot experiments, and for the *on-grid* electron microscopy assay $2 \times 10^7$ $CAP_{eq}$/condition were used.

## Preparation of cytosol

Cytosolic extracts were prepared as described before (*Radtke et al., 2010*; *Radtke et al., 2014*), dialyzed (7 K MW cut-off cassettes; Slide-A-Lyzer, Thermo Scientific), snap-frozen and stored at –80 °C. Prior to their use, the cytosols were supplemented with 1 mM ATP, 1 mM GTP, 7 mM creatine phosphate, 5 mM DTT, and PIs (Roche cOmplete), and centrifuged at 130,000 g for 30 min at 4 °C (TLA-120.2). We added nocodazole to 25 μM to the cytosols, and left them either untreated (ATP/$GTP^{high}$) or supplemented them with 10 U/mL apyrase (Sigma; ATP/$GTP^{low}$) for 15 min at RT.

## Assembly of capsid-host protein complexes *in-solution*

Capsids were resuspended in CBB and cytosol at a protein concentration of 0.2 mg/mL in an assay volume of 60 μL per sample on a rotating platform at 800 rpm for 1 hr at 37 °C (c.f. *Figure 2—figure supplement 1*). The capsid-host protein complexes were sedimented through a 30% sucrose cushion at 110,000 g for 20 min at 4 °C (TLA-100, Beckman-Coulter), resuspended in CBB by ultrasound tip sonication at 40 W for about 5 × 5 s on ice, and analyzed by mass spectrometry, immunoblot, or electron microscopy (*Radtke et al., 2014*).

## SDS-PAGE and immunoblot

The samples were lysed in Laemmli buffer (1% [w/v] SDS, 50 mM Tris-HCl, pH 6.8, 1% [v/v] β-mercaptoethanol, 5% [v/v] glycerol, bromophenol blue, PIs [Roche cOmplete]). The proteins were separated on linear 7.5% to 12% or 10% to 15% SDS-PAGE, transferred to methanol-activated PVDF membranes, probed with rabbit or murine primary antibodies (*Supplementary file 3*) and secondary antibodies conjugated to fluorescent infrared dyes (anti-rabbit IgG-IRDye1 800CW; anti-mouse IgG-IRDye1 680RD; *Supplementary file 3*) and documented with an Infrared Imaging System (Odyssey, Image Studio Lite Quantification Software, LI-COR Biosciences, Lincoln, Nebraska, USA).

## Mass spectrometry sample preparation and measurement

Capsid-host protein complexes were analyzed by liquid chromatography coupled to tandem mass spectrometry (LC-MS/MS) in four independent biological replicates. The samples were resuspended in hot Laemmli buffer and separated in NuPAGE 4% to 12% Bis-Tris protein gels (Invitrogen) before *in*-gel digestion. Briefly, proteins were fixed and stained by Coomassie solution (0.4% G250, 30%

methanol, 10% acetic acid). Sample lanes were excised, destained (50% ethanol, 25 mM ammonium bi-carbonate), dehydrated with 100% ethanol and dried using a SpeedVac centrifuge (Eppendorf, Concentrator plus). Gel pieces were rehydrated in trypsin solution (1/50 [w/w] trypsin/protein) overnight at 37 °C. Tryptic peptides were extracted in extraction buffer (3% trifluoroacetic acid, 30% acetonitrile), dried using a SpeedVac centrifuge, resuspended in 2 M Tris-HCl buffer before reduction and alkylation using 10 mM Tris(2-carboxyethyl)phosphine, 40 mM 2-Chloroacetamide in 25 mM Tris-HCl pH 8.5. The peptides were purified, concentrated on StageTips with three C18 Empore filter discs (3 M), separated on a liquid chromatography instrument, and analyzed by mass spectrometry (EASY- nLC 1200 system on an LTQ-Orbitrap XL; Thermo Fisher Scientific) as described before (*Hubel et al., 2019*). Peptides were loaded on a 20 cm reverse-phase analytical column (75 µm column diameter; ReproSil-Pur C18-AQ 1.9 µm resin; Dr. Maisch) and separated using a 120 min acetonitrile gradient. The mass spectrometer was operated in Data-Dependent Analysis mode (DDA, XCalibur software v.3.0, Thermo Fisher).

## Mass-spectrometry data analysis

Raw files were processed with MaxQuant using iBAQ quantification and Match Between Runs option, and the protein groups were filtered with Perseus for reverse identification, modification site only identification, and MaxQuant contaminant list (https://maxquant.net/maxquant/, v1.6.2.10; https://maxquant.net/perseus/, v1.6.5.0; *Cox and Mann, 2008*; *Tyanova et al., 2016a*; *Tyanova et al., 2016b*). The iBAQ intensities were normalized across all samples to the overall median intensity of the HSV-1 capsid protein VP5. Cytosol and beads incubated with cytosol samples were normalized to all proteins detected in at least three replicates in each condition. Significant differences between given conditions were determined by a two-sided Welch t-test on protein groups present in three replicates of at least one condition, followed by permutation-based FDR statistics (250 permutations), using an absolute $\log_2$ difference cut-off of 1 and an FDR cut-off of 0.05. To characterize the IFN induction, we annotated proteins reported as being induced by IFN type-I as *ISGs* proteins (InterferomeDB, > 2 x change; http://www.interferome.org/interferome/home.jspx; *Rusinova et al., 2013*). We used the Fisher's exact test against *ISGs* proteins as well as all Gene Ontology terms (GO; *Ashburner et al., 2000*; *Consortium, 2021*; http://geneontology.org/) for enrichment analysis of proteins upregulated in IFN-induced $M\varphi_{IFN}$ cytosol over $M\varphi_R$ cytosol ($\log_2$ difference ≥1.5; permutation-based FDR ≤ 0.05). The data were summarized in volcano or bar plots (GraphPad Prism v5.0, https://www.graphpad.com/; Perseus v1.6.5.0; *Tyanova et al., 2016b*).

## Interaction network assembly

We focused our analysis on proteins that showed specific differences from one capsid preparation to the other, within the same cytosol preparation, and considered host proteins with an enrichment higher than 1.5 $\log_2$ fold changes and a permutation-based FDR ≤ 0.01 as specifically enriched. To visualize enrichment among different capsid-host protein complexes, we generated integrative networks using Cytoscape (http://www.cytoscape.org/; v3.7.2; *Shannon et al., 2003*) and STRING (confidence score: 0.7; *Szklarczyk et al., 2019*). STRING uses a combination of databases on co-expression, conserved occurrences, GO terms and Kyoto Encyclopedia of Genes and Genomes (KEGG; https://www.genome.jp/kegg/; *Kanehisa and Goto, 2000*; *Kanehisa, 2019*; *Kanehisa et al., 2021*). To assemble pathway enrichments, we used DAVID, a Database for Annotation, Visualization and Integrated Discovery (https://david.ncifcrf.gov/home.jsp; v6.8; *Huang et al., 2009a*; *Huang et al., 2009b*) and the Cytoscape plug-ins ClueGO and CluePedia (http://apps.cytoscape.org/apps/cluego, v2.5.7; http://apps.cytoscape.org/apps/cluepedia, v1.5.7; *Bindea et al., 2009*; *Bindea et al., 2013*).

## Electron microscopy

Capsid-host protein complexes were assembled at ATP/GTP$^{high}$ in solution, harvested by ultracentrifugation, resuspended in CBB, and adsorbed onto enhanced hydrophilicity-400 mesh formvar- and carbon-coated copper grids (Stork Veco, The Netherlands; *Radtke et al., 2010*; *Roos et al., 2009*). Moreover, capsids at a concentration of $1 \times 10^7$ $CAP_{eq}$/mL were adsorbed directly for 20 min at RT onto the grids. The grids were incubated on a 10 µL drop of cytosol with a protein concentration of 0.2 mg/mL and ATP/GTP$^{high}$ in a humid chamber for 1 hr at 37 °C. The samples were left untreated or labeled with anti-VP5 (pAb NC-1) and protein-A gold (10 nm diameter; Cell Microscopy Centre,

Utrecht School of Medicine, The Netherlands). For both protocols, the grids were washed with PBS and ddH$_2$O, contrasted with 2% uranyl acetate at pH 4.4, air dried, and analyzed by transmission electron microscopy (Morgani or Tecnai; FEI, Einthoven, The Netherlands). The capsid morphology was evaluated for about 100 structures/assay from about 15 randomly selected images of 2.7 μm² of three biological replicates. We classified capsomer-containing structures as *punched*, if they lacked one or more of their vertices but still had an icosahedral shape, and as *flat shells*, if they lacked the icosahedral shape but contained capsomers, and scored them as one capsid equivalent structure if they contained more than 100 capsomers.

## Capsid DNA uncoating assay

D capsids were incubated with cytosols from A549-control, A549-MxB(1-715), A549-MxB(M574D), or A549-MxB-FLAG for 1 hr at 37 °C or treated for 5 min with 1% SDS followed by 10 min with 10% TX-100 (*Ojala et al., 2000*). The viral genomes released during the assay were degraded by adding 50 U/mL of benzonase for 1 hr at 37 °C, and the remaining protected DNA was purified with the DNA Blood Mini Kit (Qiagen, Hilden, Germany) and quantified by real-time PCR on a qTower³ (Analytik Jena, Jena, Germany). The SYBR Green assay was performed with the Luna Universal qPCR Master Mix (NEB, Ipswich, MA, USA) according to the manufacturer's instructions with primers specific for HSV-1 gB (UL27 gene) (HSV1_2 SYBR fwd: 5'-gtagccgtaaaacggggaca-3' and HSV1_2 SYBR rev: 5'-ccgacctc aagtacaacccc-3'; *Engelmann et al., 2008*). Standards and samples were run in triplicates and results expressed as % released viral DNA with the SDS/Tx-100 treatment normalized to 100%.

## Quantification and statistical analyses

We performed Welch's t-testing, Kruskal-Wallis H-testing, Friedman and one-way analyses of variance with a Dunns or Bonferroni post-testing (GraphPad Prism v5.0; https://www.graphpad.com/).

## Acknowledgements

We thank Katinka Döhner and Franziska Hüsers (Institute of Virology, Hannover Medical School) as well as Miriam Schilling (University of Oxford, UK) for many constructive discussions and feedback on the manuscript, and Jasper Götting (Institute of Virology, Hannover Medical School) for support on bioinformatics analyses. We are grateful to Ari Helenius (ETH Zürich, Switzerland), Graham Ogg (University of Oxford, UK), Gary Cohen (University of Pennsylvania, USA), Helena Browne (Cambridge University, UK), Jay Brown (University of Virginia, USA), Jeffrey Cohen (NIH, Bethesda, USA), Pat Spear (Northwestern Medical School, USA), and Roselyn Eisenberg (University of Pennsylvania, USA) for their generous donation of virus strains and invaluable antibodies.

Our research was supported by the EU 7th framework (Marie-Curie Actions ITN-EDGE; https://ec.europa.eu/research/mariecurieactions/about/innovative-training-networks_en, H2020-EU.1.3.1, #675,278 to JR, AP, and BS), the UK MRC (core funding of the Medical Research Council Human Immunology Unit, MC_UU_00008/8 to JR), the NIH (NIGMS, GM114141 to IMC), an EU ERC consolidator grant (ERC-CoG ProDAP 817798 to AP), the German Research Foundation (http://www.dfg.de/; PI1084/3, PI1084/4, PI1084/5, TRR179, and TRR237 to AP; KO1579/13 to GK; CRC900 C2 158989968, EXC62 REBIRTH 24102914, EXC2155 RESIST 390874280, SO403/6 to BS) and the Deutsches Zentrum für Infektionsforschung (DZIF) (TTU 07.826_00 to BS). The funders had no role in study design, data collection and analysis, decision to publish, or preparation of the manuscript.

## Additional information

### Funding

| Funder | Grant reference number | Author |
|---|---|---|
| Horizon 2020 Framework Programme | H2020-EU.1.3.1 | Jan Rehwinkel Andreas Pichlmair Beate Sodeik |
| Medical Research Council | MC_UU_00008/8 | Jan Rehwinkel |

| Funder | Grant reference number | Author |
|---|---|---|
| National Institutes of Health | NIGMS, GM114141 | Ileana M Cristea |
| European Research Council | ERC-CoG ProDAP 817798 | Andreas Pichlmair |
| Deutsche Forschungsgemeinschaft | PI1084/3 | Andreas Pichlmair |
| Deutsche Forschungsgemeinschaft | PI1084/4 | Andreas Pichlmair |
| Deutsche Forschungsgemeinschaft | PI1084/5 | Andreas Pichlmair |
| Deutsche Forschungsgemeinschaft | TRR179/TP11 | Andreas Pichlmair |
| Deutsche Forschungsgemeinschaft | TRR237/A07 | Andreas Pichlmair |
| Deutsche Forschungsgemeinschaft | KO1579/13-1 | Georg Kochs |
| Deutsche Forschungsgemeinschaft | CRC900 C2, 158989968 | Beate Sodeik |
| Deutsche Forschungsgemeinschaft | EXC62 REBIRTH, 24102914 | Beate Sodeik |
| Deutsche Forschungsgemeinschaft | EXC2155 RESIST, 390874280 | Beate Sodeik |
| Deutsche Forschungsgemeinschaft | SO403/6, 443889136 | Beate Sodeik |
| Deutsches Zentrum für Infektionsforschung | TTU 07.826_00 | Beate Sodeik |

The funders had no role in study design, data collection and interpretation, or the decision to submit the work for publication.

## Author contributions

Manutea C Serrero, Conceptualization, Data curation, Formal analysis, Investigation, Methodology, Validation, Visualization, Writing - original draft, Writing – review and editing; Virginie Girault, Data curation, Formal analysis, Investigation, Methodology, Software, Validation, Writing – review and editing; Sebastian Weigang, Data curation, Methodology, Resources, Validation, Writing – review and editing; Todd M Greco, Data curation, Formal analysis, Methodology, Software, Supervision, Writing – review and editing; Ana Ramos-Nascimento, Data curation, Formal analysis, Investigation, Writing – review and editing; Fenja Anderson, Methodology, Writing – review and editing; Antonio Piras, Data curation, Formal analysis, Methodology, Software, Writing – review and editing; Ana Hickford Martinez, Data curation, Formal analysis, Investigation, Methodology, Resources, Writing – review and editing; Jonny Hertzog, Data curation, Investigation, Methodology, Resources, Writing – review and editing; Anne Binz, Data curation, Investigation, Methodology, Resources; Anja Pohlmann, Investigation, Methodology, Resources, Writing – review and editing; Ute Prank, Methodology, Resources; Jan Rehwinkel, Funding acquisition, Methodology, Project administration, Resources, Supervision, Writing – review and editing; Rudolf Bauerfeind, Conceptualization, Data curation, Formal analysis, Investigation, Methodology, Supervision, Writing – review and editing; Ileana M Cristea, Conceptualization, Funding acquisition, Methodology, Resources, Software, Supervision, Writing – review and editing; Andreas Pichlmair, Conceptualization, Funding acquisition, Methodology, Project administration, Resources, Software, Supervision, Writing – review and editing; Georg Kochs, Conceptualization, Funding acquisition, Project administration, Resources, Supervision, Writing – review and editing; Beate Sodeik, Conceptualization, Formal analysis, Funding acquisition, Investigation, Project administration, Resources, Supervision, Writing - original draft, Writing – review and editing

Author ORCIDs
Manutea C Serrero (iD)http://orcid.org/0000-0001-8221-2725
Jonny Hertzog (iD)http://orcid.org/0000-0002-7089-982X
Andreas Pichlmair (iD)http://orcid.org/0000-0002-0166-1367
Beate Sodeik (iD)http://orcid.org/0000-0003-4650-3036

### Decision letter and Author response
Decision letter https://doi.org/10.7554/eLife.76804.sa1
Author response https://doi.org/10.7554/eLife.76804.sa2

## Additional files

### Supplementary files
• Supplementary file 1. Host proteins in THP-1 cytosols. Intensity-Based Absolute Quantitation (iBAQ) counts of the host proteins identified in the proteomic analysis of the cytosolic extracts prepared from rested or IFN-induced THP-1 φ cytosol. Statistical analyses were performed with a Welch's t-test. The following cut-offs were set for differentially-expressed proteins: permutation-based false-discovery rate (FDR) $\leq$ 0.05 and $|log_2$ fold-change$| \geq$ 1.5. The protein groups were filtered to keep only the intensities measured in at least three out of four replicates per condition. Gene Ontology knowledge was used to reference the proteins previously described as induced by interferon.

• Supplementary file 2. Host proteins in capsid-host protein complexes. Intensity-Based Absolute Quantitation (iBAQ) counts of host proteins identified in the $V_{0.1}$, $V_{0.5}$, $V_1$ and D capsid-host protein complexes assembled in rested or IFN-induced THP-1 φ cytosol. Statistical analyses were performed with a Welch's t-test. The following cut-offs were set for differentially bound proteins: permutation-based false-discovery rate (FDR) $\leq$ 0.05 and a $|log_2$ fold-change $\geq$1.5$|$. The protein groups were filtered to keep only those with intensities measured in at least three out of four replicates, in at least one condition. "Interaction significance" column indicates the proteins considered as specific interactors.

• Supplementary file 3. Viral proteins in capsid-host protein complexes. Intensity-based absolute quantification (iBAQ) counts of HSV-1(17$^+$)Lox viral proteins from isolated $V_{0.1}$, $V_{0.5}$, $V_1$ and D capsids (A) normalized to the intensity of the major capsid protein VP5, (B) unnormalized LFQ intensities. The viral proteins were filtered to keep only those with intensities measured in at least three out of four replicates, in at least one condition.

• Supplementary file 4. List of Antibodies. mAb: monoclonal antibody. pAb: polyclonal antibody. Anti-capsid SY4563 (*Döhner et al., 2018*); Anti-VP5 NC-1 (*Cohen et al., 1980*); anti-calnexin (*Hammond and Helenius, 1994*); Anti-MxA/MxB M143 (*Flohr et al., 1999*).

• Transparent reporting form

### Data availability
The raw datasets produced in this study are available at PRIDE (PXD028276; http://www.ebi.ac.uk/pride). The dataset analyses and the raw bottling images are included in the Supplementary Files 1-3 and in the Source Data folder, respectively.

The following dataset was generated:

| Author(s) | Year | Dataset title | Dataset URL | Database and Identifier |
|---|---|---|---|---|
| Serrero MC, Girault V, Pichlmair A, Sodeik B | 2021 | The interferon-inducible antiviral GTPase MxB promotes capsid disassembly and genome release of herpesviruses | https://www.ebi.ac.uk/pride/archive/projects/PXD028276 | PRIDE, PXD028276 |

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
