## [Editor Report]

This paper uses an innovative cell-free protein-protein interaction system to identify factors that interact with HSV-1 capsids in infected cells. In addition to cataloging numerous capsid-interacting proteins, the manuscript probes the antiviral mechanism of one of these, MxB. The data provide strong support for an intriguing model in which MxB "punches" holes in HSV-1 capsids, releasing viral DNA and potentially triggering host DNA sensors. Moreover, the results suggest that viral proteins bind to and shield the capsids from MxB attack, offering a new perspective on how viruses might evade some host defenses.

---

## [Decision Letter]

**Decision letter after peer review:**

Thank you for submitting your article "The interferon-inducible GTPase MxB promotes capsid disassembly and genome release of herpesviruses" for consideration by *eLife*. Your article has been reviewed by 2 peer reviewers, one of whom is a member of our Board of Reviewing Editor, and the evaluation has been overseen by Päivi Ojala as the Senior Editor. The reviewers have opted to remain anonymous.

The reviewers have discussed their reviews with one another, and the Reviewing Editor has drafted this to help you prepare a revised submission. Both reviewers were impressed with the innovative experimental design, the extensive amount of data presented, and most importantly the new insights emerging from your experiments showing how MxB might function to inhibit HSV-1 and how HSV-1 might evade the MxB defense. The reviewers felt that parts of the manuscript need important clarifications but they do not feel additional experiments are necessary. Please address the concerns listed in the specific comments below.

Essential revisions:

1) The reviewers both ended up feeling that the initial studies cataloging the large amount of mass spectrometry data in the various samples did not logically lead to the very interesting in-depth focus of MxB. Can you explain more clearly your rationale for focusing on MxB from among all the factors identified?

2) The presentation and interpretation of the data in figure 8 should be clarified to address the concerns noted by reviewer 2.

*Reviewer #1 (Recommendations for the authors):*

I just have a few suggestions on possible follow-up studies with regards to the mechanism of MxB's capsid destruction.

1. Would SDS-PAGE or Westerns with antibodies to capsid structural proteins at different stages during the MxB induced capsid degradation reveal if any of the capsid structural proteins are altered?

2. In the cell-free assay would any of the MxB extracts degrade another DNA virus capsid such as adenovirus, polyomavirus, or a phage capsid? This would be another way of showing that specific capsid surface proteins are required for MxB capsid interaction.

3. Figure S6. Would an MxB antibody label capsids in these immuno-gold EM studies?

*Reviewer #2 (Recommendations for the authors):*

The link between the first, proteomics part of the paper, and the second part, (MxB mechanism) seems tenuous. Unless I am missing it, I don't see MxB appearing in Figure S2C, Figure S3, Figure 3, or the supplementary tables. In Figure 2, MxB appears only to be slightly (insignificantly?) preferentially associated with D capsids. Can the author make a stronger rationale for linking the two parts of the paper? The proteomics data are potentially useful for future studies, but since the authors have not independently validated the findings (except by repetition), the presentation of them could be more succinct and possibly even more of the data (e.g., Figure 2) could be presented only as supplemental figures.

Line 265-268, Figure 7F. Did the authors test V[1.0] capsids? That analysis would be useful, especially since the results shown in Figure 7 and Figure 8 are not exactly consistent with each other and with the model they propose. Figure 8 shows that the viral proteins associated with the various V capsid preparations are quite similar to each other. But in Figure 7, it seems that V[0.5] capsids are more sensitive than V[0.1] capsids to damage by MxB.

Lines 292-293. The text describing the similarities and differences in proteins binding to the various capsid preparations does not fit well with the data in Figure 8. For example, V[0.1] capsids do not really contain substantially more pUL41, especially compared to V[1] capsids; V[0.1] capsids contain similar amounts of pUL40 as V[1] capsid, but much more than V[0.5] capsids; etc. Perhaps the author could focus on differences that are statistically different. These data do not identify any major proteins that show a gradient from low in D capsids to higher in [0.5] to highest in [0.1] capsids that would fit with the gradient of effects shown in Figure 7F. The discussion of these results should reflect the limitation of these results.

---

## [Author Response]

Essential revisions:1) The reviewers both ended up feeling that the initial studies cataloging the large amount of mass spectrometry data in the various samples did not logically lead to the very interesting in-depth focus of MxB. Can you explain more clearly your rationale for focusing on MxB from among all the factors identified?

We modified the text accordingly (c.f. response to reviewer 2, comment #1).

2) The presentation and interpretation of the data in figure 8 should be clarified to address the concerns noted by reviewer 2.

We modified the text accordingly (c.f. response to reviewer 2, comment #3).

Reviewer #1 (Recommendations for the authors):I just have a few suggestions on possible follow-up studies with regards to the mechanism of MxB's capsid destruction.1. Would SDS-PAGE or Westerns with antibodies to capsid structural proteins at different stages during the MxB induced capsid degradation reveal if any of the capsid structural proteins are altered?

Thank you for this suggestion. We agree that immunoblot analyses of time-course experiments on MxB induced capsid disassembly might detect consecutive dissociation of vertex-associated proteins such as the tegument proteins pUL36 and pUL37 as well as the capsid proteins pUL17 and pUL25. Although such experiments would be very informative, we agree that the required workload would exceed this revision.

Nevertheless, some considerations from our side. We are quite satisfied that we could transfer our experiments from analyzing sedimented capsids and capsid disassembly intermediates to the on-grid analyses (c.f. Figure 6A; Figure 7E, F, G).

To detect several of the structural HSV-1 proteins in one lane in an immunoblot, we required capsids derived from about 1 to 2 x 175 cm^2^ cell culture flasks, for other HSV-1 proteins we would need even more. This is about 10 times as many capsids as for3 replicates in the on-grid EM assay (c.f. manuscript, lines 222-224). Thus, such immunoblot experiments are quite demanding in terms of working hours and consumables. Nevertheless, we plan to try the suggested experiments, either by immunoelectron microscopy or by SDS-PAGE.

Moreover, we treasure the “open-view” of the EM analysis. After we had discovered MxB-induced capsid disassembly, we became concerned whether disassembly intermediates might be less likely to sediment than intact capsids, and whether disassembly intermediates might be further damaged by ultracentrifugation and more importantly by the subsequent re-suspension.

2. In the cell-free assay would any of the MxB extracts degrade another DNA virus capsid such as adenovirus, polyomavirus, or a phage capsid? This would be another way of showing that specific capsid surface proteins are required for MxB capsid interaction.

Previous work from Georg Kochs et al. (Schilling et al. 2018) as well as Jovan Pavlovic et al. (Crameri et al. 2018) shows that MxB fails to restrict several RNA viruses, adenovirus Ad5, or vaccinia virus indicating that MxB shows antiviral activity against specific viruses.

Accordingly, we could not detect MxB binding to Ad5 capsids in our co-sedimentation assays (unpublished observations). In preliminary experiments with papillomavirus-like particles, we detected an MxB induced exposure of an internal epitope by immunoelectron microscopy (unpublished observations). Furthermore, we are expanding our studies to HCMV, MCMV and MHV68 capsids. While these experiments will be informative, we think that they require more work, and that they are beyond the scope of the present manuscript.

3. Figure S6. Would an MxB antibody label capsids in these immuno-gold EM studies?

Although such experiments would be informative, we think that they require more work, and that they are beyond the scope of the present manuscript.

So far, the commercially MxB antibodies that we tested did not label capsids treated with MxB cytosol in our immunoelectron microscopy trials. Therefore, we plan additional experiments with FLAG-tagged MxB and anti-FLAG antibodies.

Reviewer #2 (Recommendations for the authors):The link between the first, proteomics part of the paper, and the second part, (MxB mechanism) seems tenuous. Can the author make a stronger rationale for linking the two parts of the paper?

In this study, we designed our cell-free capsid-host interaction assays (reviewed in Radtke *et al.* 2014) to identify IFN inducible proteins potentially restricting HSV-1 capsids, and therefore focused on proteins of the GO terms “*Response to type I IFN”* and “*Regulation of type I IFN production”* (c.f. Figure 4) binding to capsids.

To answer this comment, we have expanded the explanation for our rationale to focus on MxB (previous lines 179 – 183; in the revised manuscript lines 181 – 189):

“Particularly interesting was the discovery of MxB in these capsid-host protein complexes. MxB was significantly enriched on HSV-1 D capsids in Mφ_IFN_ cytosol, and the IFN treatment had the strongest impact on the interaction of MxB with capsids. Moreover, the calculated enrichment score for MxB on capsids was very high although the MxB levels in the input cytosol were below the detection limit (undetected, Figure S2, Table S1). MxB but not its homolog MxA restricts infections of the herpesviruses HSV-1, HSV-2, MCMV, KSHV, and MHV-68, but its mode of action has not been elucidated (Crameri et al. 2018; Liu et al. 2012; Schilling et al. 2018; Vasudevan et al. 2018). For these reasons, we investigated the interaction of human MxB with HSV-1 capsids further.”

Unless I am missing it, I don't see MxB appearing in Figure S2C, Figure S3, Figure 3, or the supplementary tables. In Figure 2, MxB appears only to be slightly (insignificantly?) preferentially associated with D capsids.

In addition to line 85 of the introduction, we added to line 180: “…. were enriched for IFI16, OAS2, POLR1C, STAT2 and MxB (gene Mx2) in Mφ_IFN_ but not in Mφ_R_ (Figure 4, Figure S5).” to clearly indicate its protein name and its gene name.

Figure S2C reports on the changes in the protein composition of the cytosolic extracts upon IFN treatment. In these samples, the MxB induction was so low compared to the other IFN inducible proteins that it was not detected by mass spectrometry. In Figure S3B, MxB is listed under its gene name MX2 in bold in the 4^th^ column as one protein of the 4^th^ cluster; it is number 8 from the bottom of the top block of the 4^th^ column. MxB is one of the many ISG proteins that we detected on the capsids, and that we indicated with the # symbol in this graph. In the supplementary tables, MxB is listed under its gene name MX2.

In Figure 2, we indicated in red those ISG proteins that were significantly enriched (permutation based FDR ≤ 0.05; log_2_ difference ≥ 1.5) on either capsid type in these binary comparisons, and MxB enriched on D capsids in comparison to the 3 other viral capsid types.

In our functional enrichment analyses we used a conservative significance cutoff of an enrichment higher than 1.5 log_2_ (2.83-fold change) and a permutation-based FDR ≤ 0.01 for the GO and the STRING analyses (Figure S4 and Figure 3). MxB did not reach these cut-offs and is therefore not represented in these graphs.

The proteomics data are potentially useful for future studies, but since the authors have not independently validated the findings (except by repetition), the presentation of them could be more succinct and possibly even more of the data (e.g., Figure 2) could be presented only as supplemental figures.

It is true that so far we have validated only the MxB interaction with capsids in depth. But Figure 2 provides a comparative overview on the complexity of the macrophage host protein-capsid complexes. We consider this dataset valuable and would prefer to present it without the need to consult supplementary figures and table.

Figure 2 shows that different protein sets bound to different viral capsids (0.1, 0.5 and 1 M) in comparison to the D capsids (compare A and B versus C as well as D and E versus F). As the tegumentation of the 0.1 M capsids was more complex, the set of associated host proteins was also more complex. Moreover, the capsids associate with other proteins in the IFN cytosol than in the control cytosol (compare left panels with respective right panels).

Therefore, we would like to keep Figure 2 among the main figures; also in light of comment 2.

Line 265-268, Figure 7F. Did the authors test V[1.0] capsids? That analysis would be useful, especially since the results shown in Figure 7 and Figure 8 are not exactly consistent with each other and with the model they propose. Figure 8 shows that the viral proteins associated with the various V capsid preparations are quite similar to each other. But in Figure 7, it seems that V[0.5] capsids are more sensitive than V[0.1] capsids to damage by MxB.

Unfortunately, we did not compare V[1.0] capsids in a time course disassembly experiment with V[0.5], V[0.1], or D capsids. We think that V[0.5] capsids are most similar to authentic cytosolic capsids, which unfortunately we cannot isolate in sufficient amounts for biochemical experiments.

V[0.5] capsids are the best substrates to recruit microtubule motors (Radtke et al. 2010), to move along microtubules in vitro (Wolfstein et al. 2006), dock to NPCs (Ojala et al. 2010; Anderson et al. 2014), and to release their genomes (Ojala et al. 2000). Thus, although the mass spectrometry analysis revealed only a few clear changes in the protein composition of V[1.0] capsids, V[0.5] capsids, and V[0.1] capsids, the functional assays suggest that the conformation of the HSV-1 proteins exposed on the capsid surfaces differ.

Based on these other studies, we therefore hypothesize that the differences in macrophage proteins associating with V[0.1] versus V[0.5] versus V[1] capsids were due to protein denaturation rather than protein extraction or removal from the capsids by increasing the KCl concentration from 0.1 to 0.5 to 1 M.

Lines 292-293. The text describing the similarities and differences in proteins binding to the various capsid preparations does not fit well with the data in Figure 8. For example, V[0.1] capsids do not really contain substantially more pUL41, especially compared to V[1] capsids; V[0.1] capsids contain similar amounts of pUL40 as V[1] capsid, but much more than V[0.5] capsids; etc. Perhaps the author could focus on differences that are statistically different. These data do not identify any major proteins that show a gradient from low in D capsids to higher in [0.5] to highest in [0.1] capsids that would fit with the gradient of effects shown in Figure 7F. The discussion of these results should reflect the limitation of these results.

We agree with the reviewer that we detected very few overall differences in the protein composition of V[0.5] when compared to V[1] capsids. The D capsids were most different from the 3 other capsid types. Furthermore, the major differences of V[0.1] versus V[0.5/1] capsids were in pUS3, VP13/14 and pUL16.

We therefore hypothesize that the differences in macrophage proteins associating with V[0.5] versus V[1] capsids were due to protein denaturation rather than protein extraction or removal from the capsids by increasing the KCl concentration from 0.5 to 1 M.

We have therefore modified lines 290-291 of the Results section:

“In contrast, V_0.1_ capsids contained more tegument proteins, e.g. pUS3, VP13/14 and pUL16. All capsid preparations contained traces of membrane proteins and nuclear HSV-1 proteins contributing to DNA replication and packaging (Figure S8).”

Furthermore, we expanded lines 312-314 (now lines 309-320) of the discussion:

“While our MS analyses showed that V_0.5_ and V_1_ capsids recruited unique but also common proteins, the proteomes of V_0.1_ and D capsids were more distinct. These specific interactions are consistent with the notion that a treatment with 0.5 or 1 M KCl during the detergent lysis of virions destabilized intra-tegument interactions that influenced, for example, the recruitment of dynactin, kinesin-1 and kinesin-2 from brain cytosol (Ojala et al., 2000; Radtke et al., 2010; Wolfstein et al., 2006).

Moreover, these results are consistent with immunoelectron microscopy data showing that the surface of distinct V capsid types display different tegument epitopes (Radtke *et al.*, 2010), and with cryoelectron tomography data revealing diminishing tegument densities from V_0.1_, V_0.5_, V_1_ capsids to C capsids (Anderson *et al.*, 2014). Thus, the surface features of V_0.1_, V_0.5_ and V_1_ capsids differ as indicated by cryoelectron tomography, binding of anti-tegument antibodies, and the recruitment of distinct sets of cytosolic proteins from brain tissue (Radtke *et al.*, 2010), or macrophages as shown here.”